# 16S rRNA gene amplicon-based metagenomic analysis of bacterial communities in the rhizospheres of selected mangrove species from Mida Creek and Gazi Bay, Kenya

Edith M. Muwawa[1,2]*, Chinedu C. Obieze[3], Huxley M. Makonde[4], Joyce M. Jefwa[1,5], James H. P. Kahindi[1], Damase P. Khasa[2]

1 Department of Biological Sciences, Pwani University, Kilifi, Kenya, 2 Centre for Forest Research and Institute for Systems and Integrative Biology, Laval University, Québec, Canada, 3 Africa Centre of Excellence in Oilfield Chemicals Research, University of Port Harcourt, Choba, Nigeria, 4 Department of Pure & Applied Sciences, Technical University of Mombasa, Mombasa, Kenya, 5 National Museums of Kenya, Nairobi, Kenya

* edithmuwawa@yahoo.com

**Data Availability Statement:** All relevant data are within the manuscript and its Supporting Information files. Sequences from this study have

## Abstract

Prokaryotic communities play key roles in biogeochemical transformation and cycling of nutrients in the productive mangrove ecosystem. In this study, the vertical distribution of rhizosphere bacteria was evaluated by profiling the bacterial diversity and community structure in the rhizospheres of four mangrove species (*Sonneratia alba*, *Rhizophora mucronata*, *Ceriops tagal* and *Avicennia marina*) from Mida Creek and Gazi Bay, Kenya, using DNA-metabarcoding. Alpha diversity was not significantly different between sites, but, significantly higher in the rhizospheres of *S. alba* and *R. mucronata* in Gazi Bay than in Mida Creek. Chemical parameters of the mangrove sediments significantly correlated inversely with alpha diversity metrics. The bacterial community structure was significantly differentiated by geographical location, mangrove species and sampling depth, however, differences in mangrove species and sediment chemical parameters explained more the variation in bacterial community structure. Proteobacteria (mainly *Deltaproteobacteria* and *Gammaproteobacteria*) was the dominant phylum while the families *Desulfobacteraceae*, *Pirellulaceae* and *Syntrophobacteraceae* were dominant in both study sites and across all mangrove species. Constrained redundancy analysis indicated that calcium, potassium, magnesium, electrical conductivity, pH, nitrogen, sodium, carbon and salinity contributed significantly to the species–environment relationship. Predicted functional profiling using PICRUSt2 revealed that pathways for sulfur and carbon metabolism were significantly enriched in Gazi Bay than Mida Creek. Overall, the results indicate that bacterial community composition and their potential function are influenced by mangrove species and a fluctuating influx of nutrients in the mangrove ecosystems of Gazi Bay and Mida Creek.

been deposited into the SRA under the BioProject ID PRJNA644929.

**Funding:** This research was funded by the Queen Elizabeth II Jubilee Scholarship to EMM and NSERC discovery grant to DPK and the International Foundation for Science (Ref No.: IFS Grant A/5719-1) to HMM. The funder had no role in study design, data collection and analysis, decision to publish, or preparation of the manuscript.

**Competing interests:** The authors have declared that no competing interests exist.

## Introduction

Mangroves are coastal wetland forests mainly found at the intertidal zones of estuaries, brackish waters, deltas, creeks, lagoons, marshes and mudflats of tropical and subtropical latitudes [1]. Globally mangroves are mainly distributed in Asia (42%), Africa (20%), North and Central America (15%), Oceania (12%) and South America (11%) and cover approximately 60% to 75% of the world's tropical and subtropical coastlines [2, 3].

The mangrove ecosystems provide important ecological and economic functions, including protecting coastlines from storm damage and erosion, degrading environmental contaminants, and providing nursery habitats for numerous aquatic organisms [4]. In addition, they form an enormous food web, providing a myriad of microorganisms with nutrients [5]. The regular tidal variations, pH, temperature, salinity, light, rainfall and nutrient availability create a dynamic environment within the mangrove ecosystems [6], which provides a niche for a wide range of organisms. Thus, the complex mangrove ecosystems have generated increased interest among microbial ecologists [7]. Complex interactions of the microbes in plant ecosystems maintain the harmony of different biogeochemical processes and sustain the nutritional status and ecological balance [8]. Special environmental conditions such as high salinity, high sulfur and low oxygen, drives the proliferation of unique microorganisms that contribute to sulfate reduction, methane cycling and ammonia oxidation within mangrove sediments [9]. Therefore, it is important to understand the bacterial structure and species composition underlining mangrove sediments, particularly within the rhizospheres of specific mangrove species.

A recent study by Wu et al. [10] on the vertical distributions of rhizosphere bacteria from three mangrove species in Beilun Estuary, South China showed that bacterial communities from mangrove rhizosphere sediments were dominated by Proteobacteria (mostly *Delta*-proteobacteria and *Gamma*-proteobacteria) and to some extent Chloroflexi, Bacteroidetes, Planctomycetes and Acidobacteria. Furthermore, their study revealed that plant species and depth played a role in shaping the microbial communities, however, the influence of deterministic factors (environmental parameters) such as pollution and mangrove encroachment were not considered. Considering the important ecological functions of mangrove microbiomes and the fact that this unique ecosystem lies at the interface between land and ocean, investigating the deterministic factors that shape microbial communities in line with increasing climate change is crucial.

Previous studies on mangroves in Kenya have concentrated on floristic composition and distribution of mangrove species, economic utilization and regeneration strategies of the principal species [11]. However, data on microbial community diversity is limited due to low attention in exploring the mangrove habitats for microbial diversity [12, 13], an information gap that this study has also addressed. The few microbial diversity studies that have been conducted were mainly culture-dependent [12]. Considering that about 98% of microbes are uncultivable, the 'omic' technologies are emerging as potential tools to ensure holistic insight into environmental systems [14] and this has led to the increasing application of high-throughput sequencing in studying microbial communities [10, 15]. In this study, the vertical profiles of bacteria communities in the rhizospheres of four mangrove species namely, *Sonneratia alba*, *Rhizophora mucronata*, *Ceriops tagal* and *Avicennia marina*, were determined and compared. Since we anticipated some possible contamination at Mida Creek attributed to human-induced activities such as firewood harvesting, pollution from plastics and faeces, pollution from oil spills, overharvesting for building materials and encroachment for settlements [16, 17], we obtained our samples from two distinct mangrove ecosystems; viz, a pristine site in Gazi Bay, located in the South Coast of Kenya and the polluted site in Mida Creek, located in

the North Coast of Kenya. We successfully applied for the first-time in-depth analysis of the bacterial communities within the rhizospheres of four mangrove species from Kenya, using high-throughput sequencing of the V3–V4 regions of the 16S rRNA gene on Illumina MiSeq platform.

## Materials and methods

### Ethical statement

The National Commission for Science, Technology and Innovation of Kenya (NACOSTI) approved this research study, National Environmental Management Authority of Kenya (NEMA) provided the access permit (for field sampling), Kenya Wild life Services (KWS) and Kenya Plant Health Inspectorate Services (KEPHIS) provided permits that facilitated the shipment of samples to Laval University, Canada. The field studies neither involved endangered nor protected species.

### Study site

Two study sites (Mida Creek and Gazi Bay) in Kenya were investigated. Mida Creek is located in Kilifi County (03°34'S, 039°96'E), about 88 km North of Mombasa and approximately 25 km south of Malindi town in a planigraphic area of 32 km$^2$ [18]. It is characterized by a hot and humid tropical climate with an average annual temperature of 27°C. Humidity is high throughout the year, up to 90% relative humidity during the rainy season [18]. In addition, the creek is affected by anthropogenic degradation such as overharvesting of mangroves for firewood, timber and fish traps, pollution from plastics, faeces and oil spills, clearing of mangrove and conversion to other land uses such as aquaculture, urban development and tourism [16, 17].

Gazi Bay is located in Kwale County (04°44'S 039°51'E), approximately 55 Km from Mombasa, South Coast of Kenya. The Bay is sheltered from strong waves by the presence of the Chale peninsula to the East and a fringing coral reef to the South. The climate is hot and humid. The average annual temperature and humidity are about 28°C and upto 95%, respectively [18]. The mangrove forests in both sites display a similar zonation and mangrove species contribution among the four dominant species *A. marina* (30%), *R. Mucronata* (25%), *S. alba* (15%) and *C. tagal* (10%), which contribute about 80% of the total mangrove formation in both forests. *Sonneratia alba* (about 6–10 m tall) forms the outermost zone (seaward side) towards the open water followed by pure stands of *Rhizophora mucronata* (about 8–12 m tall) or mixed stands of *Rhizophora mucronata* and *Bruguiera gymnorrhiza* (about 10–20 m tall) and in turn these stands are followed by pure stands of *Ceriops tagal* (about 3–5 m tall) and *Avicennia marina* (about 12–18 m tall) along the creek [19–21]. *Rhizophora mucronata* have well-developed prop roots that accumulate large stocks of debris, perhaps contributing to some accretion that supports the extensive tidal flats seen in the area [21].

### Collection of samples

Sampling was conducted in May 2018 according to the described methods by Wu et al. [10]. During sampling, both sites had low tides with temperatures of 27°C and 28°C for Mida Creek and Gazi Bay, respectively. From pure stands of *S. alba*, *R. mucronata*, *C. tagal* and *A. marina*, one tree was selected per stand. The selected trees were spaced at least 10 m apart. For each individual mangrove species, four points (~2 meters distance apart) were selected for sampling. The sampling was done at two depths (1–5 cm and 10–15 cm), using a standardized core sampler [22]. The coring was conducted closely to the mangrove roots in order to capture the rhizosphere. A total of 64 samples (4x4x2x2) were kept in sterile plastic bags, maintained

in a dry iced box before they were transported and stored at -20˚C before further analyses was performed at the Laval University, Canada.

## Nutrient analysis of sediment samples

Nutrient analyses of soil samples for nitrogen, carbon, phosphorus, potassium, calcium, magnesium and sodium were conducted according to standard methods [23]. Determination of pH and electrical conductivity was done using the calcium chloride method at a ratio of 1:2 using a digital pH meter [Corning pH meter 140] [24] and electrical conductivity meter [Conductivity meter type CDM 2d radiometer Copenagen] [25], respectively.

## Total community DNA extraction, PCR protocol and Illumina MiSeq sequencing

The excised roots containing rhizosphere sediment were placed in a 50 mL sterile falcon tube containing autoclaved phosphate buffer and shaken for 2 min to release the rhizosphere sediment from the surface of the roots. Total genomic DNA was extracted directly from 0.25 g of the sediment using Power Soil DNA isolation kit (DNeasy PowerSoil Kit, Qiagen, Germany) in accordance with the manufacturer's protocol. The extracted DNA was quantified using NanoDrop 1000 spectrophotometer (Thermo Scientific, USA). For each PCR reaction, 20 ng of total DNA was used as a template. Briefly, amplification of the bacterial 16S rRNA gene fragment (V3-V4 region) was performed using the sequence specific regions described in [26]. The following generic oligonucleotide sequences were used for amplification: Bakt_341F-long `AATGATACGGCGACCACCGAGATCTACAC`[index1]`TCGTCGGCAGCGTCAGATGTGTATAA GAGACAGCCTACGGGNGGCWGCAG` and Bakt_805R-long `CAAGCAGAAGACGGCATACGAGAT`[index2]`GTCTCGTGGGCTCGGAGATGTGTATAAGAGACAGGACTACHVGGGTATCTAATCC`. The primers used in this work contain Illumina specific sequences protected by intellectual property (Oligonucleotide sequences © 2007–2013 Illumina, Inc., all rights reserved). The PCR was carried out in a total volume of 50 μL that contains 1X Q5 buffer (NEB), 0.25 μM of each primer, 200 μM of each dNTPs, 1 U of Q5 High-Fidelity DNA polymerase (NEB) and 1 μL of template DNA. The PCR started with an initial denaturation at 98˚C for 30s followed by 10 cycles of denaturation at 98˚C for 10s, annealing at 55˚C for 10s, extension at 72˚C for 30s. An additional 25 cycles were as follows; denaturation at 98˚C for 10s, annealing at 65˚C for 10s, extension at 72˚C for 30s, followed by a final extension step at 72˚C for 2 min. The PCR amplicons was purified using the Axygen PCR cleanup kit (Axygen). Quality of the purified PCR product were checked on a DNA7500 BioAnalyzer chip (Agilent) and quantified using a Nanodrop 1000 (Thermo Fisher Scientific). Barcoded Amplicons were pooled in equimolar concentration and sequenced on the Illumina Miseq (paired-end 300 bases with two index reads) at the Institute for Integrative System Biology (IBIS)/ (Laval University's Genomic Analysis Platform, Quebec, Canada). Sequences from this study have been deposited into the SRA under the BioProject ID PRJNA644929.

**Sequence processing and functional prediction.** Illumina MiSeq sequences were processed using QIIME2 v2018.11 [27]. Chimeric sequences, marginal sequence errors and noisy sequences were filtered while picking amplicon sequence variants (ASVs) using DADA2 [28]. Further binning of the ASVs at 97% similarity into Operational Taxonomic Units (OTUs) was done using VSEARCH open-reference OTU picking strategy [29] against the SILVA v132 [30] reference database. Representative sequences were assigned taxonomy using a trained Naïve Bayes classifier (SILVA 132) for 16S rRNA V3 –V4 hyper variable region using the q2-feature-classifier plugin. Singletons were removed and the OTU count table was rarefied prior to the

determination of alpha and beta-diversity using *ampvis2* (v2.5.5) [31] and the *vegan* v2.5–6 [32] R packages [33].

Predictive functional analysis of bacterial communities was done using Phylogenetic Investigation of Communities by Reconstruction of Unobserved States 2 (PICRUSt2). To predict functions, representative sequences were first aligned to HMMER [34]. The alignment was then placed into a reference tree using EPA-NG [35] and gappa [36]. Multiple 16S rRNA gene copies were then normalized and gene families predicted using Castor–a hidden state prediction tool [37]. The predicted gene families were subsequently collapsed into MetaCyc pathways using MinPath [38].

**Statistical analyses.** Unless otherwise stated, statistical analyses were performed using R v3.6.1 [33]. A three-factor (sites, mangrove species and depth differences) test of differences in α-diversity and physicochemical parameters was done by the non-parametric Kruskal-Wallis H test using the *agricolae* [39] R package. Post hoc test for mean separations was based on Fisher's least significant difference. Comparison of microbial community differences (β-diversity) in the two study sites, among mangrove species and between sampling depths were based on Bray-Curtis dissimilarities and principal coordinates analysis (PCoA) using *ampvis2* and *ape* [40] R packages. Permutational multivariate analysis of variance (PERMANOVA) was used to test the significance of the differences in multivariate space using the *vegan* package. Furthermore, a post-hoc test on significant PERMANOVA ($q \leq 0.05$) was performed using the "adonis_pairwise" function in the *metagMisc* R package. The influence of physicochemical parameters on community-level differentiation (community structure) was determined by constrained redundancy analysis (RDA). RDA was performed on Hellinger-transformed bacterial and environmental data using a combination of *ampvis2* and *vegan* R package functions. The significance of the constraining variables was determined based on permutation test using the *vegan* function 'anova.cca'. Further investigation on the contribution of environmental factors on the microbial community differentiation was achieved by variance partitioning (based on chi-square) using the *vegan* R package.

Differential abundance (Benjamini-Hochberg false discovery rate (FDR)-adjusted, $p \leq 0.05$) of OTUs in both sites of study and among mangrove species was determined using *aldex2* [41] R package. Linear Discriminant Analysis Effect Size (LEfSe) [42] was used to determine the presence of biomarker PICRUSt2 predicted KEGG orthologs (enzymes) by applying the Kruskal–Wallis alpha significance threshold of $\leq 0.01$ and an LDA (linear discriminant analysis) score of 3.0. The GraPhlan software [43] was subsequently used to visualize the detected biomarkers. Pathways significant differentiation was determined by also applying the Kruskal–Wallis alpha significance threshold of $\leq 0.05$ and FDR-adjusted Fisher's least significant difference post hoc analysis using the *agricolae* R package.

# Results

## Sequencing statistics, diversity measures and richness estimates

A total of 3,038,718 sequence reads were obtained from the 64 mangrove rhizosphere samples, out of which 189,254 high-quality 16S rRNA sequence reads were clustered into 4,295 OTUs. The rarefaction of counts to 1,126 reads per-sample was sufficient for explaining differences in bacterial diversity (S1 Fig). Approximately 56% of the OTUs were shared between Gazi Bay and Mida Creek. Also, mangrove plant species comparison revealed that *A. marina* and *C. tagal* had the most OTUs (1341 and 1332, respectively). *Sonneratia alba* comprised 978 OTUs while *R. mucronata* had the least number of OTUs (901). Comparison of shared OTUs in the two study sites revealed that 24.3% of the core OTUs (OTUs found in $\geq$ 20% of the samples from both sites) were shared between the rhizospheres of *A. marina* in Gazi Bay and Mida

Creek (S2A Fig). For *S. alba*, 22.6% of the core OTUs were shared between the two sites while 9.8% and 4.6% of the core OTUs were respectively shared in the rhizosphere of *R. mucronata* and *C. tagal* for Gazi Bay and Mida Creek. Overall, 12.8% of the core OTUs were shared between the two sites (S2B Fig).

Bacterial diversity in the rhizosphere of the mangrove plant species revealed significant differences on comparison of the two sites investigated (Table 1). Shannon-Wiener, Observed and Chao1 diversity indices were all significantly higher (Fisher's LSD, $p \leq 0.01$) in the rhizosphere of *S. alba* in Giza bay compared to Mida Creek. Other rhizosphere species comparison in Giza bay and Mida Creek were not significantly different apart from Shannon-Wiener diversity for *A. marina*, which was found to be significantly higher ($p \leq 0.05$) in Mida Creek (Table 1). Overall, α-diversity was significantly higher ($p \leq 0.001$) in the rhizosphere of *A. marina* in both Giza bay and Mida Creek compared to the rhizosphere of other plant species (S3A Fig), while depth of sampling had no significant ($p > 0.05$) influence on α-diversity differentiation (S3B Fig) within mangrove species in each site and across both study sites. Further consideration of alpha diversity pattern based on horizontal sampling (mangrove zonation) revealed that species diversity and richness was lower in the seaward lower tidal zones (*S. alba* and *R. mucronata*) compared to the upper tidal zones (*A. marina* and *C. tagal*). Also, an overall comparison of α-diversity without considering differences in plant species, revealed that there were no significant differences ($p > 0.05$) between Giza bay and Mida Creek (S3A Fig).

Spearman's rank correlation of α-diversity measures to physicochemical parameters revealed significant correlations. Ca ($r = -0.3$, $p = 0.01$), C ($r = -0.38$, $p = 0.001$), Mg ($r = -0.39$, $p = 0.001$), N ($r = -0.24$, $p = 0.05$) and K ($r = -0.34$, $p = 0.006$) all significantly correlated negatively with Observed species, while salinity, pH, Na, EC and phosphorus did not correlate significantly. For Shannon-Wiener measure of diversity, all the physicochemical parameters showed significant negative correlation except for pH, which did not correlate significantly (S1 Table). The correlation results of Chao1 to physicochemical properties was similar to the Observed as both matrices measures specie richness. For Chao1, Ca ($r = -0.31$, $p = 0.01$), C ($r = -0.37$, $p = 0.002$), Mg ($r = -0.38$, $p = 0.001$) and K ($r = -0.33$, $p = 0.007$) were all negatively correlated to species richness, while salinity, pH, Na, EC and phosphorus did not correlate significantly. Overall, only pH failed to correlate with any of the α-diversity matrices.

## Bacterial composition and differences across sites and mangrove species

The bacterial composition of Gazi Bay and Mida Creek was found to be highly diverse, comprising a total of 25 bacterial phyla with at least 1% abundance in the samples. *Proteobacteria* (41%) was the most abundant phylum in both sites. This was followed by *Planctomycetes*

**Table 1. Mean alpha diversity measures for the different plant species in Mida Creek and Giza bay.**

| Diversity measure | *A. marina* | | *C. tagal* | | *R. mucronata* | | *S. alba* | |
|---|---|---|---|---|---|---|---|---|
| Sites | Gazi | Mida | Gazi | Mida | Gazi | Mida | Gazi | Mida |
| Shannon-Wiener | 4.40 ± 0.22 | 4.65 ± 0.13* | 4.36 ± 0.16 | 4.29 ± 0.29 | 3.70 ± 0.32 | 3.83 ± 0.38 | 4.38 ± 0.20*** | 3.67 ± 0.17 |
| Chao1 | 138.61 ± 14.95 | 158.19 ± 26.09 | 130.32 ± 18.08 | 125.91 ± 33.14 | 85.96 ± 15.15 | 87.03 ± 17.62 | 123.87 ± 21.98*** | 70.93 ± 11.14 |
| Observed | 132.12 ± 16.03 | 147.62 ± 21.13 | 120.87 ± 14.86 | 118.50 ± 27.49 | 80.75 ± 16.35 | 84.00 ± 15.93 | 117.00 ± 21.64*** | 69.13 ± 8.97 |

Values represent mean and standard deviation. Superscripts beside values are significantly different measures ($p \leq 0.05$) based on Fisher's least significance difference. (Significance codes: 0

'***' 0.001

'**' 0.01

'*' 0.05)

(12%), Bacteroidetes (7%) *Acidobacteria* (7%), *Chloroflexi* (6%), *Epsilonbacteraeota* (5%), *Actinobacteria* (3%), *Gemmatimonadetes* (2%), *Cyanobacteria* (2%), *Latescibacteria* (2%) and *Firmicutes* (2%). Other phyla with relative abundance >1% are presented in (Fig 1A). Eight phyla were found to be differentially abundant between Gazi Bay and Mida Creek. *Dadabacteria* (FDR-corrected $p = 0.00$), *Nitrospirae* ($p = 0.04$), *Planctomycetes* ($p = 0.01$) and *Rokubacteria* ($p = 0.04$) were all differentially abundant in Gazi Bay while *Entotheonellaeota* ($p = 0.01$), *Modulibacteria* ($p = 0.00$), *Spirochaetes* ($p = 0.00$) and *Gemmatimonadetes* ($p = 0.01$) were differentially abundant in Mida Creek. Comparison of mangrove species across both sites indicated that differentially abundant phyla for *C. tagal* were Dadabacteria ($p = 0.05$), Rokubacteria ($p = 0.003$) and Verrucomicrobia ($p = 0.01$). Only Cyanobacteria ($p = 0.05$) was differentially abundant on comparison of *R. mucronata* in Gazi Bay and Mida Creek while there were no differentially abundant phyla on comparison of both sites for *A. marina* and *S. alba*. Depth-based comparison of the bacterial composition within mangrove species in individual sites and across sites did not reveal any differentially ($p > 0.05$) abundant phyla.

At the family taxonomic level, 80.2% of the species were successfully classified. *Desulfobacteraceae* (9%), *Pirellulaceae* (6%), *Syntrophobacteraceae* (4%) *Thiovulaceae* (3%) and *Geminicoccaceae* (2%) were among the most abundant families across Gazi Bay and Mida Creek (S4 Fig). No bacterial families were found to be differentially abundant between Gazi Bay and Mida Creek, however, there were significant differences in the rhizosphere of plant species. *Balneolaceae, Geminicoccaceae, Gemmatimonadaceae, Entotheonellaceae, Nitrospiraceae, Nitrosococcaceae, Phycisphaeraceae, Pirellulaceae* and *Kiloniellaceae* were all differentially enriched ($p \leq 0.05$) in the rhizosphere of *A. marina*. In the rhizosphere of *R. mucronata*,

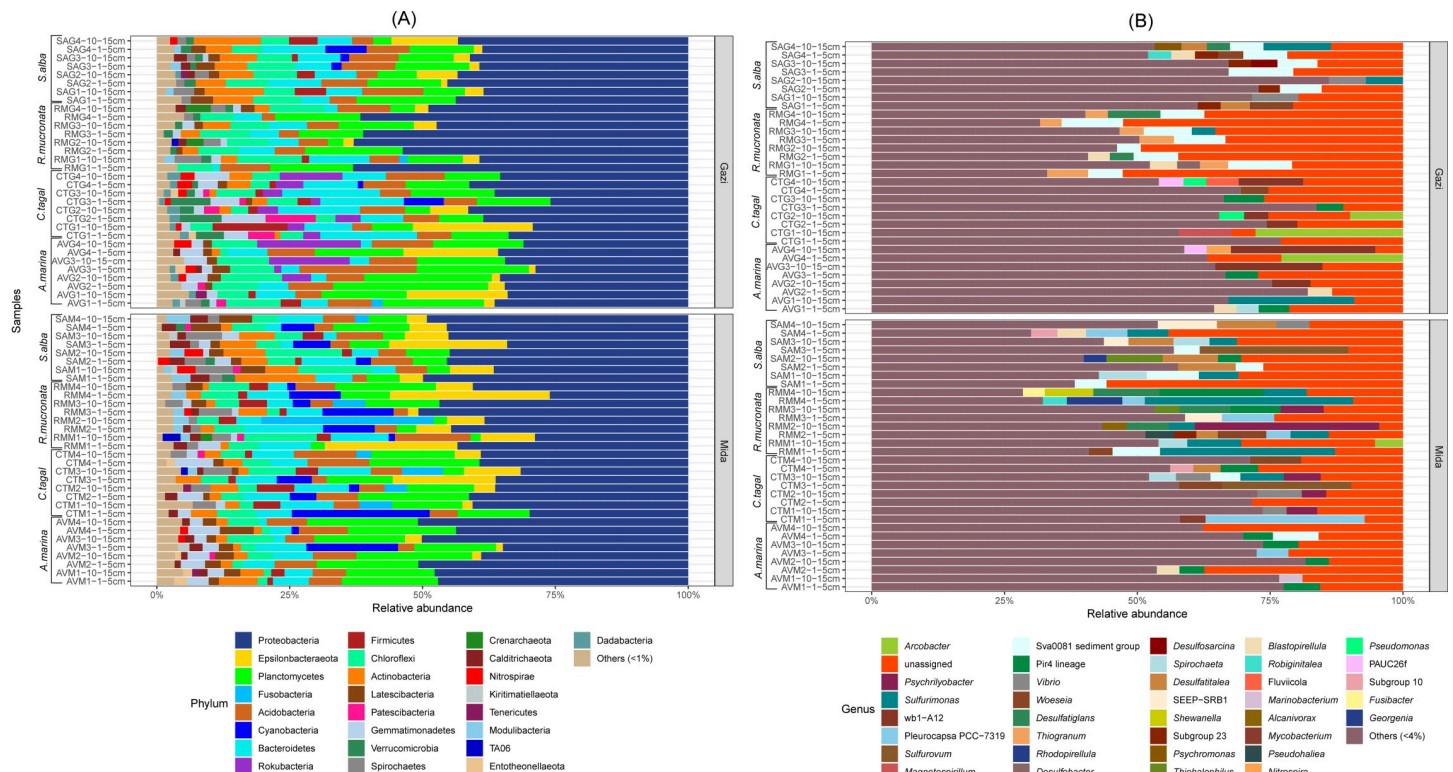

**Fig 1.** (A) Phyla level-based characterization of the rhizosphere sediments. (B) Genus level characterization of the sites of investigation showing only genera with at least 4% relative abundance in any on the samples. *Bacterial genera with abundance less than 4% collectively comprised 59% of the bacterial composition at the genus level.

*Desulfobulbaceae*, *Desulfarculaceae*, *Ectothiorhodospiraceae* and *Thiotrichaceae* were differentially enriched ($p \leq 0.05$). Differentially abundant families in the rhizosphere of *S. alba* were *Flavobacteriaceae*, *Moduliflexaceae*, *Spirochaetaceae*, *Kineosporiaceae* and *Chthoniobacteraceae* while *Desulfobacteraceae* was enriched ($p \leq 0.05$) in the rhizospheres of both *R. mucronata* and *S. alba*. Overall, there were no significant differences ($p > 0.05$) in bacterial families between these two-mangrove species.

Several species were not classified (35.06%) at the genus level using the SILVA reference database. Among the top 10 classified bacterial genera across all sites and plant species were *Sulfurimonas* > Sva0081 sediment group > Pir4 lineage > *Arcobacter* > *Woeseia* > *Blastopirellula* > *Spirochaeta* > wb1-A12 > *Psychrilyobacter* > *Desulfatitalea*, according to sequence of abundance (Fig 1B). Investigation of the bacterial genera for differential abundance between mangrove species in Gazi Bay and those of Mida Creek revealed significant differences. *Nitrospira* ($p = 0.04$), uncultured *Desulfobacteraceae* ($p = 0.05$), uncultured *Sandaracinaceae* ($p = 0.04$), uncultured *Syntrophobacteraceae* ($p = 0.05$) and wb1-A12 (*Methylomirabilaceae*) ($p = 0.01$) were differentially abundant for *C. tagal* in Gazi Bay and Mida Creek. Also, uncultured *Syntrophobacteraceae* ($p = 0.01$) and *Chthoniobacter* ($p = 0.05$) were differentially abundant in the rhizosphere of *R. mucronata* in Gazi Bay and Mida Creek while an uncultured *Rhodospirillales* ($p = 0.05$) was the only bacterial genera differentially abundant for *S. alba* in Gazi Bay and Mida Creek. There were also significant differences among the different mangrove plant species. For instance, *Nitrospira*, *Blastopirellula* and *Defluviicoccus* were differentially abundant ($p \leq 0.05$) in the rhizosphere of *A. marina*. *Chthoniobacter*, *Desulfatiglans* and *Thiogranum* were differentially enriched ($p \leq 0.05$) in the rhizosphere of *R. mucronata* while *Spirochaeta* was differentially enriched ($p < 0.05$) in the rhizosphere of *S. alba*. Consideration of compositional differences based on vertical sampling did not reveal any differentially ($p > 0.05$) abundant genera (S2 Table).

## Bacterial community differentiation based on site, depth and mangrove species

Beta-diversity based on Bray-Curtis distance revealed a significant clustering of samples within the ordination space. Samples clustered mainly based on mangrove species, and much less, based on the site differences and on depth of sampling (Fig 2). The differences observed in multivariate space for mangrove species (PERMANOVA $R^2 = 16\%$, $p = 0.001$), site of sampling (PERMANOVA $R^2 = 4.5\%$, $p = 0.001$) and depth of sampling (PERMANOVA $R^2 = 4.0\%$, $p = 0.02$) were all significant. Pair-wise post hoc analysis of PERMANOVA revealed that the bacterial community composition and structure are significantly different between the mangrove plant species (FDR-adjusted $p = 0.001$) as well as between the sites ($p = 0.001$) and sampling depth ($p = 0.017$). The bacterial community composition and structural differences in the mangrove species rhizospheres is presented in details in S3 and S4 Tables.

## Site and mangrove species differences in physicochemical parameters

Site-based comparison of the different mangrove trees were mostly significant, apart from salinity and electrical conductivity that were not influenced by geographical distance for *S. alba* and *A. marina* (Table 2). pH and calcium were significantly higher (Kruskal-Wallis, $p \leq 0.05$) in all mangrove plant species of Mida Creek compared to Giza bay. Apart from the rhizosphere of *R. mucronata*, all other mangrove species rhizosphere in Giza bay had significantly lower ($p < 0.05$) physicochemical properties compared to the species in Mida Creek. The physicochemical parameters that were higher in the rhizosphere of *R. mucronata* in Gazi Bay included potassium, sodium, phosphorus, total carbon, nitrogen, salinity and electrical conductivity.

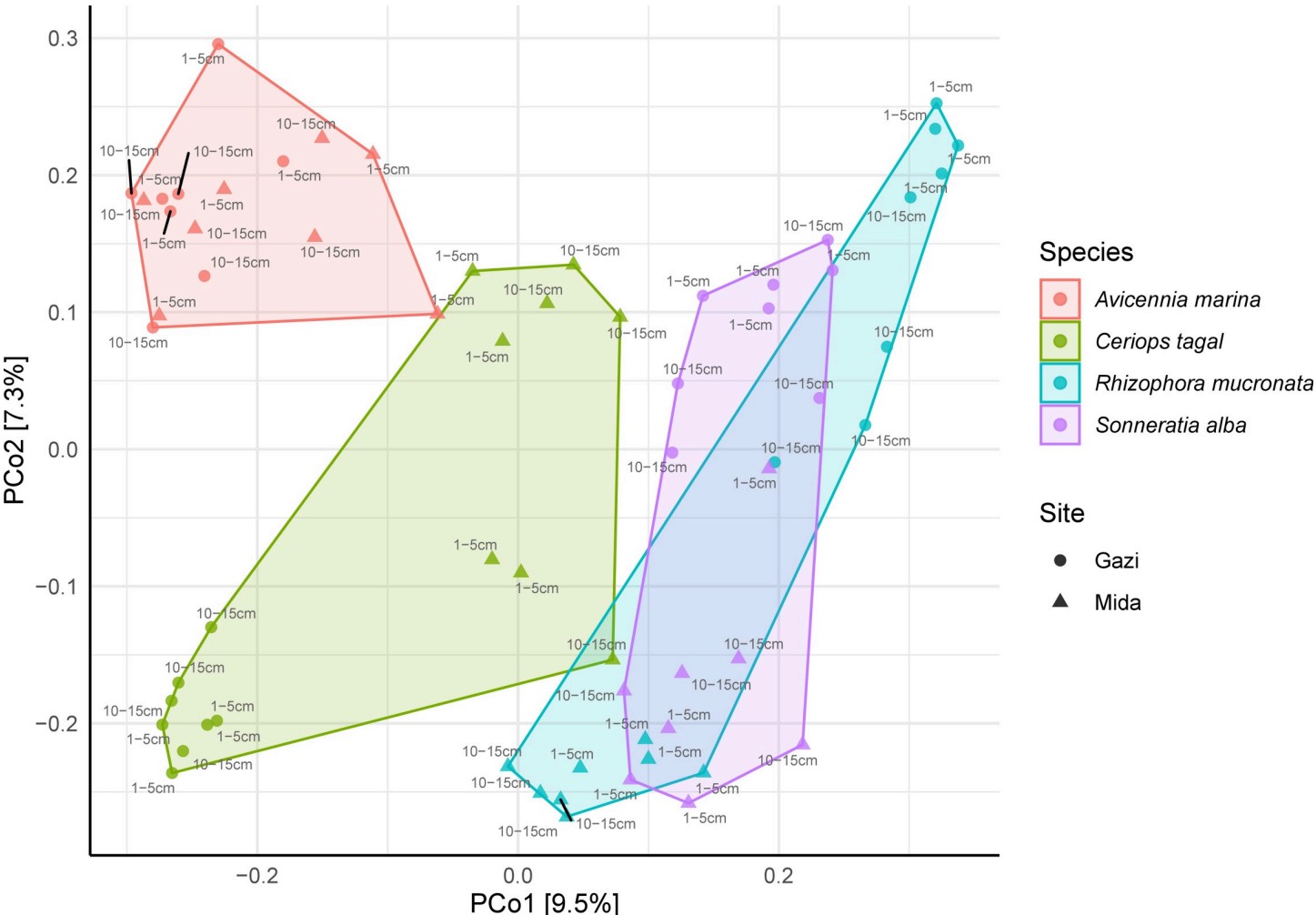

**Fig 2. Principal coordinates analysis (PCoA) of Bray-Curtis dissimilarity between the bacterial communities in different mangrove rhizospheres in Mida Creek and Gazi Bay.**

### Influence of environmental factors on bacterial composition and structure

Redundancy analysis led to the explanation of 91% of the observed variation in β-diversity out of which 18% was explained by the constrained relation of site differences, mangrove species differences and depth of sampling, while 72% was explained by the unconstrained environmental variables (Fig 3A). The total explained variation of the microbial community composition of the first two axes (RDA1 and RDA2) reached 51%. The constrained variation resulting from the differences in site, species and depth were 6.5% and 3.7% for RDA1 and RDA2, respectively. The subjection of RDA to statistical test revealed that the observed differentiation was significant ($p = 0.001$). On the overall, the significance of environmental terms fitted into the RDA model indicated that calcium, potassium, magnesium, electrical conductivity, pH and salinity with significance values of $p = 0.001$ and carbon with significance of $p = 0.002$ contributed the most to the species–environment relationship (Table 3). Other contributing physicochemical factors were nitrogen ($p = 0.007$) and sodium ($p = 0.01$), while phosphorus ($p = 0.07$) was the only physicochemical factor without a significant influence on the bacterial community. Though pH influenced the bacterial community across sites, its effect also separated the samples

**Table 2. Mean values of sediment physicochemical parameters among mangrove species in Mida Creek and Gazi Bay.**

| Physicochemical parameters | A. marina | | C. tagal | | R. mucronata | | S. alba | |
|---|---|---|---|---|---|---|---|---|
| Site | Gazi | Mida | Gazi | Mida | Gazi | Mida | Gazi | Mida |
| Calcium (mg/kg) | 325.75± 64.73 | 14085.5± 3686.55*** | 168.25± 27.76 | 79312.88 ±38869.51*** | 2670.62 ±544.23 | 62731.12 ±26956.45*** | 477± 147.65 | 52097± 18982.24*** |
| Potassium (mg/kg) | 663.12± 192.89 | 494±182.67 | 178.25± 40.78 | 464± 111.51*** | 1588.87 ±172.27*** | 597.375 ± 211.89 | 461.87± 138.12 | 594.75 ± 82.25* |
| Magnesium (mg/kg) | 379.125± 76.07 | 841.75± 470.97* | 115.12 ± 21.66 | 1776± 660.54*** | 1856.12 ±140.19* | 1307.12 ± 665.24 | 406.87± 68.52 | 1166.62± 345.31*** |
| Sodium (mg/kg) | 2369 ±815.28 | 4162±3483.99 | 155.87±40.50 | 3933.87 ±1448.40*** | 8472.25 ±967.11*** | 3263.75±1446.15 | 1978 ±431.28 | 2948.37 ±471.96*** |
| Phosphorus (mg/kg) | 55±16.29 | 136.12 ±33.09*** | 40±2.87 | 164.5±61.01*** | 187.75±49.67** | 98.87±54.49 | 55.125 ±14.06 | 99.75±36.32** |
| Total carbon (mg/kg) | 0.82±0.41 | 1.90±0.57*** | 0.26±0.07 | 4.55±1.56*** | 7.40±0.85*** | 2.69±1.67 | 0.99±0.23 | 2.25±0.60*** |
| Nitrogen (mg/kg) | 0.04±0.02 | 0.16±0.03*** | 0.01±0.01 | 0.18±0.04*** | 0.37±0.06*** | 0.11±0.05 | 0.07±0.01 | 0.09±0.01* |
| Electrical conductivity (S/m) | 5.78±0.95 | 4.56±1.39 | 3.37±1.31 | 6.51±2.98* | 12.37±1.00*** | 4.05±1.36 | 4.88±0.88 | 5.22±1.34 |
| pH | 7.05±0.55 | 8.17±0.21*** | 6.21±0.10 | 7.93±0.07*** | 6.09±0.15 | 8.08±0.28*** | 6.08±0.08 | 7.91±0.16*** |
| Salinity (mg/kg) | 3.13±0.55 | 2.43±0.79 | 1.77±0.72 | 3.58±1.75* | 7.08±0.61*** | 2.15±0.78 | 2.61±0.50 | 2.81±0.77 |

Values represent mean and standard deviation. Superscripts beside values are significantly different measures ($p \leq 0.05$) based on Fisher's least significance difference. (Significance codes: 0

'***' 0.001

'**' 0.01

'*' 0.05).

from the different sites (Gazi Bay and Mida Creek) along the y-axis (Fig 3A). *Desulfobacteraceae* and *Syntrophobacteraceae* were projected in the positive direction of RDA1 and thus observed to be positively correlated with carbon, nitrogen, potassium and magnesium, while *Ardenticatenales* appeared not to be strongly influenced by physicochemical factors.

To further disentangle the respective contribution of plant species, chemical parameters, depth and geographical variation in explaining the bacterial community structure, variance partitioning was performed (Fig 3B). The sediment chemical parameters had the highest influence on the bacterial community compared to mangrove species and site/depth. The sediment chemical parameters alone explained for 6% and in combination with other factors for a total of 13% of the bacterial community pattern followed by mangrove species (alone: 4% while in combination with other factors: 10%) and site/depth (alone: 1% while in combination with other factors: 4%). Eighty-two percent of the differences in bacterial community structure was not explained by a combination of all the measured variables, indicating that other underlying factors not investigated are also influencing the community differences. Spearman's rank correlation of physicochemical parameters to the top bacterial genera revealed significant associations (S5 Table). Pir4 lineage, *Desulfobacter*, Pleurocapsa PCC.7319, *Rhodopirellula*, *Vibrio*, *Desulfatiglans* and *Marinobacterium* correlated the most to physicochemical terms.

## Predictive functional analysis of bacterial communities

A total of 8819 PICRUSt2 predicted KEGG orthologs (enzymes) were collapsed into 406 Meta-Cyc pathways. Special focus was given to enzymes associated with energy metabolism (nitrogen metabolism, carbon fixation, methane metabolism, sulfur metabolism and photosynthesis) and the biosynthesis of important secondary metabolites. One hundred and

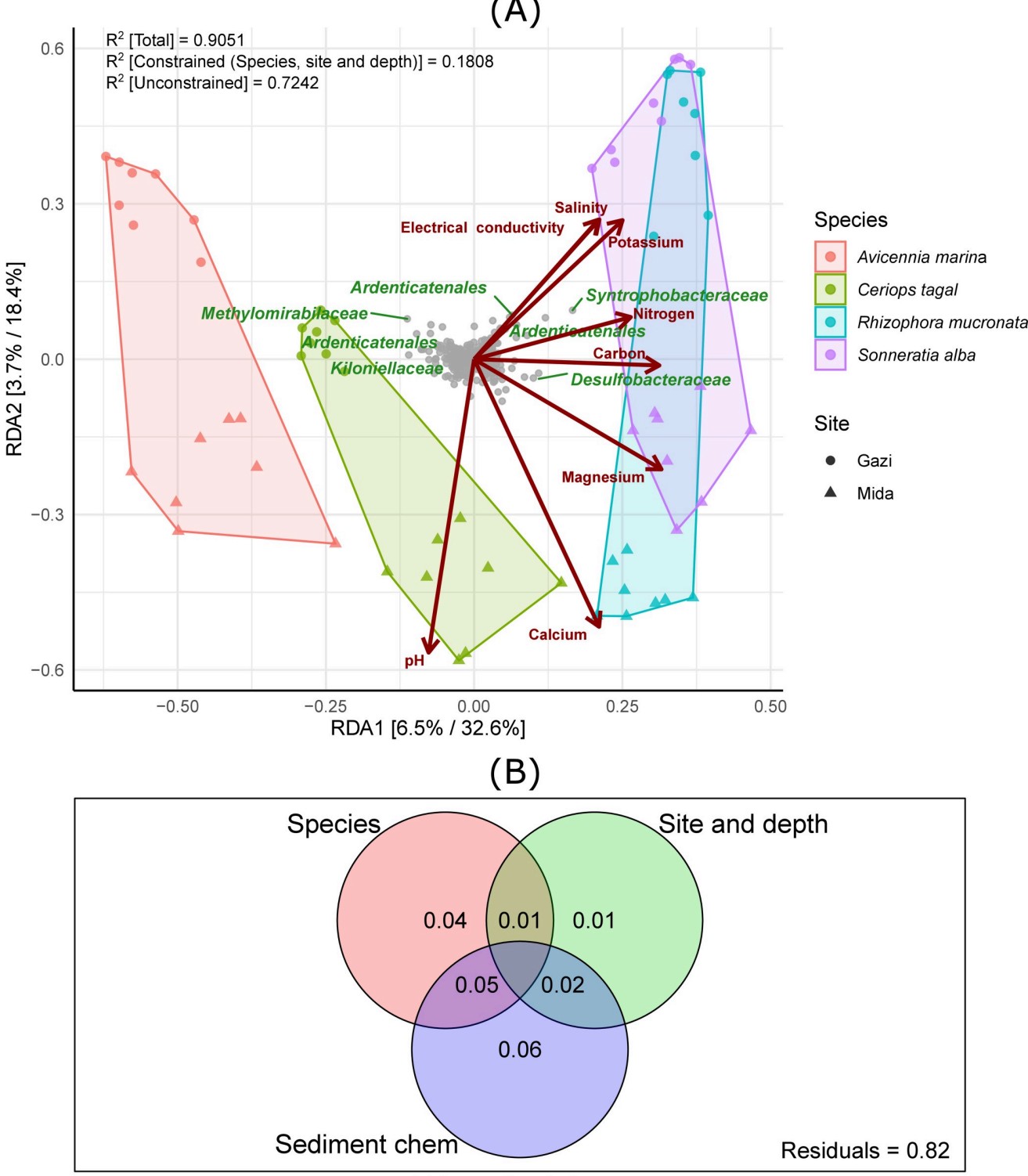

**Fig 3.** Constrained redundancy analysis displaying contributions of environmental factors to bacterial community composition (A) and (B) variation in the bacterial community structure explained by species, site/depth differences and physicochemical parameters (Ca, Mg, N, P, K, EC, salinity, pH, Na and C). *Note only environmental variables with p-value of at least 0.01 are displayed.

Table 3. Goodness-of-fit statistics (R²) for environmental factors fitted to the constrained redundancy analysis (RDA).

| Chemical parameters | RDA1 | RDA2 | r2 | Pr(>r) |
|---|---|---|---|---|
| Calcium | 0.37769 | -0.92593 | 0.6353 | 0.001*** |
| Potassium | 0.68184 | 0.7315 | 0.2748 | 0.001*** |
| Magnesium | 0.8296 | -0.55836 | 0.296 | 0.001*** |
| Sodium | 0.93518 | 0.35417 | 0.1177 | 0.019* |
| Phosphorus | 0.76524 | -0.64375 | 0.0863 | 0.067 |
| Carbon | 0.99926 | -0.03853 | 0.1993 | 0.002** |
| Nitrogen | 0.95578 | 0.29409 | 0.1559 | 0.007** |
| Electrical_conductivity | 0.61391 | 0.78937 | 0.2376 | 0.001*** |
| pH | -0.13432 | -0.99094 | 0.6669 | 0.001*** |
| Salinity | 0.61651 | 0.78735 | 0.2395 | 0.001*** |

(Significance codes: 0

'***' 0.001

'**' 0.01

'*' 0.05)

six KEGG orthologs were differentially abundant among mangrove species in Gazi Bay, whereas, 55 KOs were differentially abundant among mangrove species of Mida Creek (S6 and S7 Tables). Presented in Fig 4 are the top KOs (based on LDA score) associated with selected biogeochemical processes that are important for mangrove ecosystem productivity. Predicted KOs associated with nitrogen metabolism and photosynthesis were differentially abundant in the rhizosphere of *S. alba* species in both study sites, however, majority of the mangrove

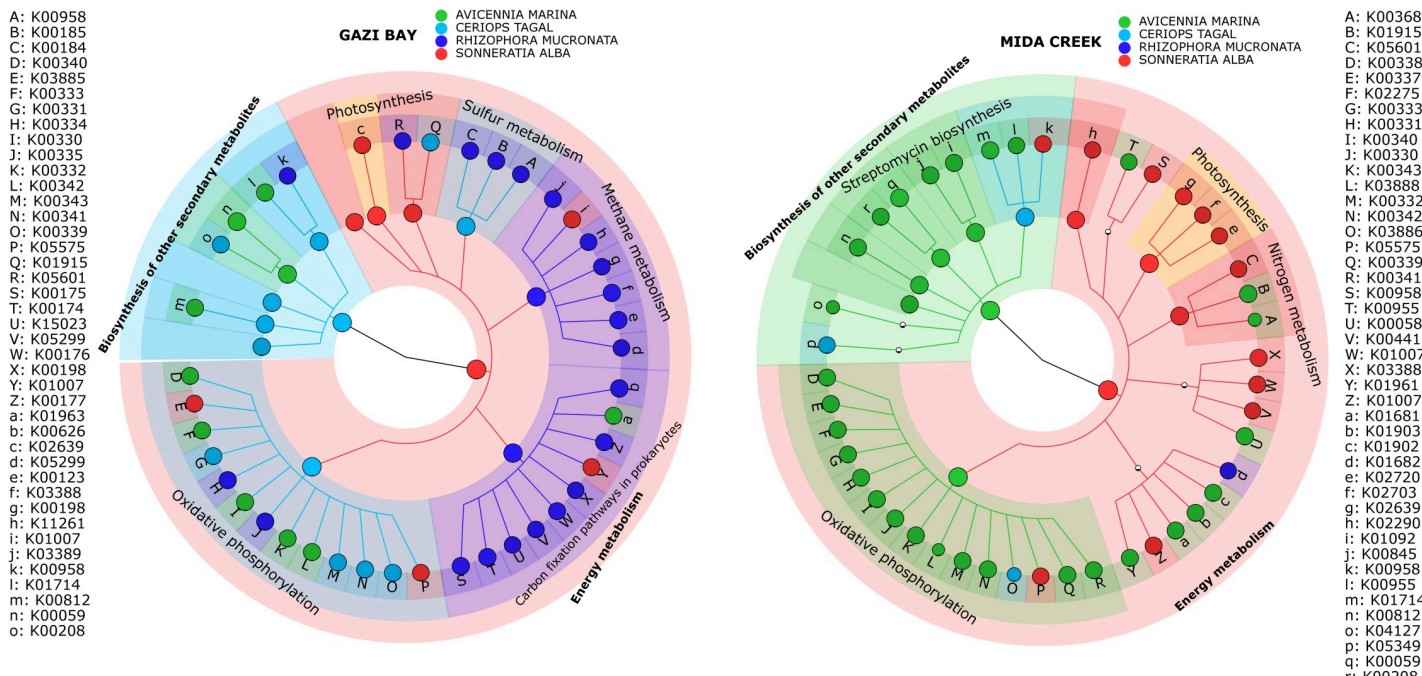

**Fig 4. Detected biomarkers for energy metabolism and the biosynthesis of important biomolecules.** Node colours distinguish between mangrove species, the background coloration differentiates the KEGG functions (KEGG level 2 and level 3 functions) while the node size corresponds to the relative abundance of the predicted KOs. Rings from the inner to the outer layer represents level 2 and 3 KEGG functions respectively.

species did not displayed uniform KEGG functional traits across sites. Based on PICRUSt2 prediction of OTUs contribution, the bacterial families *Microcystaceae*, *Coleofasciculaceae* and *Xenococcaceae* were the potential contributors to photosynthesis while *Nitrincolaceae* (*Marinobacterium*), *Pirellulaceae* (*Blastopirellula*), *Desulfobacteraceae* (*Desulfosarcina*), *Solibacteraceae*, *Defluviitaleaceae*, *Prolixibacteraceae* (*Roseimarinus*), *Entotheonellaceae*, *Ruminococcaceae*, *Geminicoccaceae*, *Thermoanaerobaculaceae*, *Phycisphaeraceae*, and *Kiloniellaceae* were the phylotypes that potentially contributed to nitrogen metabolism.

KEGG orthologs associated with carbon metabolism in prokaryotes were differentially enriched in the rhizosphere of *R. mucronata* in Gazi Bay. *Desulfatiglans*, *Vibrio*, *Desulfovibrio*, *Desulfatitalea* and *Woeseia* were the potential phylotypes contributing to carbon metabolism. Investigation of genes for methane metabolism also revealed that *R. mucronata* associated microbiomes were enriched with KOs for methane metabolism in Gazi Bay. Potential contributing phylotypes included *Desulfobacteraceae* (*Desulfosarcina*), *Desulfarculaceae* (*Desulfatiglans*) uncultured *Dehalococcoidia*, *Moduliflexaceae*, uncultured *Crenarchaeota*, *Syntrophobacteraceae*, *Thalassospiraceae* (*Thalassospira*), *Pirellulaceae*, *Psychromonadaceae* (*Psychromonas*) and *Bogoriellaceae* (*Georgenia*).

Furthermore, detected KOs associated with the synthesis of biomolecules included genes for streptomycin biosynthesis, novobiocin biosynthesis, prodigiosin biosynthesis and phenazine biosynthesis. *Pirellulaceae* (*Rhodopirellula*), *Phycisphaeraceae*, uncultured *Anaerolineae*, *Nitrosococcaceae*, *Woeseiaceae*, *Solibacteraceae*, *Methylomirabilaceae*, *Prolixibacteraceae*, *Kiloniellaceae*, *Magnetospirillaceae* (*Magnetospirillum*) and *Arcobacteraceae* we predicted to potentially contribute to the synthesis of biomolecules. Overall, the abundance of predicted KOs for sulphur metabolism, carbon metabolism and methane metabolism were higher in Gazi Bay compared to Mida Creek.

Pathway differentiation according to site and mangrove species also indicated that they were significant differences in the abundance of predicted pathways for sulfur, carbon, nitrogen and methane metabolism (Fig 5). It was observed that pathways for both sulfur and carbon metabolism were significantly higher (Fisher's LSD, *p<0.05*) in Gazi Bay than Mida Creek while other difference in predicted pathways were mangrove species-dependent. Meanwhile, for all mangrove species, depth-based comparison within site and across sites indicated that there were no significant (Fisher's LSD, *p>0.05*) differences.

## Discussion

Mangrove ecosystems are widespread in estuarine and coastal regions of the tropics, providing protection and nutrients to several microbial communities [5]. The dynamic environment within mangrove ecosystems provides an excellent niche for a wide range of organisms. In this study, we determined the bacterial communities of rhizospheric sediments from four mangrove species (*Sonneratia alba*, *Rhizophora mucronata*, *Ceriops tagal* and *Avicennia marina*), which are common along the Coastline of Kenya, using the Illunima MiSeq sequencing technology. Also, the influence of mangrove species, sampling depth, geographical location and other physicochemical factors, in shaping bacterial communities in the mangrove ecosystem was determined.

An important finding drawn from the comprehensive study of the mangroves in Gazi Bay and Mida Creek is that the different mangrove species and chemical properties had a greater influence on the rhizosphere microbiome diversity compared to sampling depth. For instance, despite the differences in geographical location, *A. marina* in both Gazi Bay and Mida Creek had the most bacterial diversity and richness (Table 1). This finding is consistent with the study by Wu et al. [10], where they observed that differences in mangrove species (*Bruguiera*

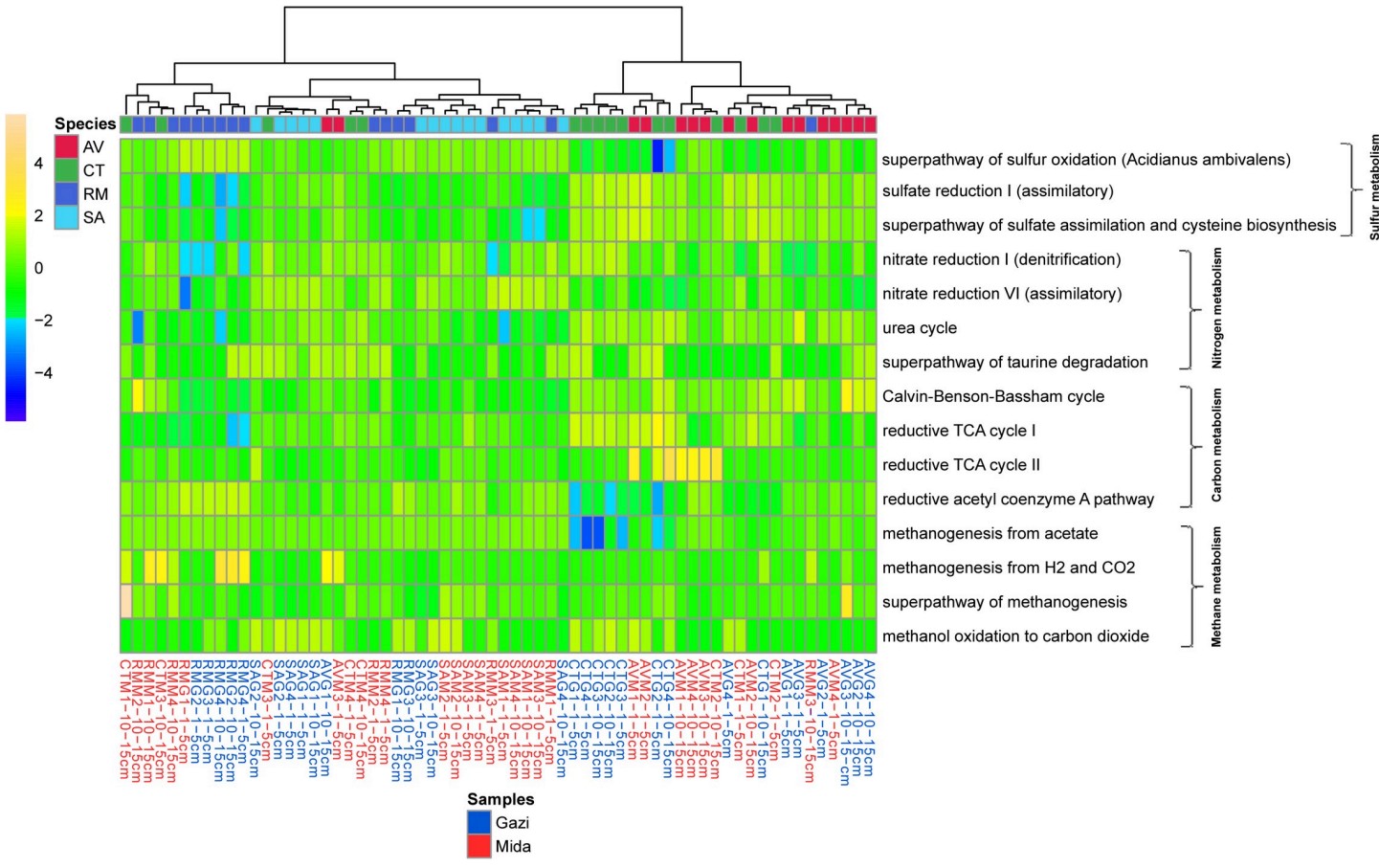

**Fig 5. Differences in predicted pathways for sulfur, carbon, nitrogen and methane metabolism in both sites and according to mangrove species.** 'AV' denotes *A. marina*, 'CT' denotes *C. tagal*, 'RM' denotes *R. mucronata*, and 'SA' denotes *S. alba*.

*gymnorrhiza*, *Kandelia candel* and *Aegiceras cornicu-latum*) had a strong pattern on bacterial α-diversity. Besides, it has been demonstrated that plant roots are capable of imposing a selective force on rhizospheric microbiomes in both mangrove swamps [44] and other terrestrial environments [45], and that this phenomenon is usually plant species dependent. As earlier stated, another important variable influencing α-diversity were the sediment chemical parameters, and this finding is consistent with previous studies of bacterial communities in mangrove ecosystems [46, 47]. In this study, most of the investigated physicochemical parameters correlated inversely with both species diversity and richness matrices (S1 Table); suggesting that an influx of nutrients from anthropogenic sources reduces bacterial diversity and the influence of mangrove species on mangrove sediment microbiomes. Notable examples were the significantly higher diversity and species richness among some mangrove species in Gazi Bay (considered pristine) compared to Mida Creek where nutrient inputs were observed to be significantly higher. Also, the low percentage of shared OTUs between same mangrove species in the two study sights (S2 Fig) highlights the importance of deterministic factors in shaping mangrove sediment microbiomes. Overall, this pattern of bacterial response indicates that in Gazi Bay and Mida Creek, the bacterial alpha diversity pattern is determined by an interaction of both mangrove species and physicochemical factors than with sampling depth.

Though differences in sampling depth did not have a strong pattern on α-diversity, investigation of bacterial community composition revealed it significantly contributed in shaping the

bacterial community (Fig 2). The clustering of samples in the PCoA plot according to mangrove species, geographical location and depth of sampling were all significant and respectively explained 16%, 4.5% and 4% of the total variation in bacterial community composition (S3 Table). In an exploratory study of the prokaryotic communities in mangrove sediments of southern China, Zhang et al. [47] indicated that there were significant interactions between sampling site and plant species that shaped the bacterial community. Similarly, Wu et al. [10] demonstrated that differences in mangrove plant species imposed a strong selective pattern on the rhizosphere microbial communities compared to depth, which did not show a clear clustering pattern, while further evidence on the influence of geographical location on mangrove microbial community composition has been presented in recent studies [48, 49].

RDA revealed a strong bacterial species–environmental variable relationship that influenced the community composition. Also, variable partitioning revealed the importance of the inter-relationship among environmental properties and between bacterial species (Fig 3B). In this study, Calcium, potassium, magnesium, electrical conductivity, pH and salinity influenced the bacteria community composition the most (Table 3). These environmental variables have been reported as factors that influence bacterial community composition and structure [15, 47, 50]. Zhang et al. [47] reported that total organic carbon and mean annual precipitation were the main environmental variables influencing the bacterial community composition of mangrove sediments in Southeastern China.

Although evidence abounds on the effect of sediment chemical parameters, mangrove species and sampling depth on bacterial communities, it is important to state that diversity and assembly patterns for this study also followed known intertidal zonation patterns. For example, alpha diversity reduced in the seawards tidal zone compared to the upper intertidal zones. Also, regardless of site differences, samples drawn from the lower intertidal zones appeared to cluster together in the PCoA ordination space. This observation is consistent with previous studies [51, 52] and thus highlights the importance of mangrove forest intertidal zonation on bacterial alpha diversity.

The phylum Proteobacteria was the most abundant in the rhizosphere of all mangrove species in both sites. Also, a total of 25 phyla had relative abundance above 1% (Fig 1A). This observed high number of bacterial phylotypes is one of the trademark features used in distinguishing the microbial communities of mangrove sediments from other biomes [47]. The dominant phyla from this study have been consistently detected in similar studies that evaluated mangrove rhizosphere bacterial communities [48, 53–55]. The high abundance of Proteobacteria can be traced to the dominant physicochemical parameters in mangrove sediments, particularly the nitrogen deficiency and the usually rich sulfur and carbon content. Dominant proteobacteria classes such as *Delta*-proteobacteria are reported to be metabolically versatile, and have a strong affinity to saline environments [56]. Besides, they actively participate in organic matter decomposition, ammonia oxidation and sulfate reduction [47]. The phylum Planctomycetes has been reported to be a component of microbial core in the mangrove ecosystems; taking part in methane and sulphur metabolism [57]. Also, Cyanobacteria, Chloroflexi and Planctomycetes are reportedly involved in nitrogen cycling in mangrove ecosystems [58]. Related studies by Yamada et al. [59] and Ghosh and Bhadury [60] also demonstrates additional importance of Chloroflexi in the decomposition of organic matter in mangrove ecosystems.

The most abundant genus, *Sulfurimonas* (Fig 1B) has been detected and recovered mostly from sulfur rich marine environments [7, 52]. Members of the genus *Sulfuriomonas* are versatile hydrogen and sulfur oxidizing chemolitotrophs [61, 62] and have been detected abundantly in both high [50] and low [52] tidal sediments as well as in hydrothermal vents [61]. Other abundant genera including *Sva0081* sediment group (Desulfobacteraceae), *Pir4 lineage*

(Pirellulaceae), *Arcobacter*, *Woeseia*, *Blastopirellula*, *Spirochaeta*, *wb1-A12* (Methylomirabilaceae), *Psychrilyobacter* and *Desulfatitalea* are also associated with sulfur rich environments and some tend to co-occur. For instance, *Sva0081*, *Woeseia* and *Spirochaeta* were recently detected among differentially enriched bacterial genera in Tanmen harbor, east coast of Hainan Island, China [63]. *Sva0081*, *Pir4 lineage*, *Spirochaeta* and *Desulfatitalea* are established sulfur oxidizers [63–65] and their abundance in the samples can be linked to the rich sulfur and carbon content of the study sites. *Pir4 lineage* and *Spirochaeta* have also been reported to contribute to the decomposition of organic matter in marine environments [65, 66]. Besides participating is sulfur oxidation, *Desulfatitalea* have been reported to also possess the *nrfA* gene for dissimilatory nitrate reduction to ammonium [64], while both *Arcobacter* and *Desulfatitalea* are suspected to play a role in the degradation of microplastics in marine environments [67]. The versatility of these organisms suggests the reason for their high abundance in the sites of study, and underlines their potential importance in pollution mitigation and biogeochemical cycling in mangrove sediments. The differentially abundant phyla and genera detected in the same mangrove species from different sites can be described as species sensitive to changing environmental factors. These sensitive bacterial species can be of importance in evaluating the impact of anthropogenic factors or climate change on mangrove microbiomes and their corresponding ecological function. Overall, the bacterial composition of Mida Creek and Gazi Bay suggests a bacterial interaction that leads to a coupled cycling of sulfur, carbon and nitrogen.

The prediction of bacterial functional profiles focused on key enzymes and pathways associated with energy metabolism and the biosynthesis of important secondary metabolites (Fig 4). The results demonstrated the functional potential of the mangrove rhizospheres bacterial communities. Oxyphotobacteria, which was predicted to contribute to photosynthesis are established primary producers in marine environments and are known to also play a role in nitrogen fixation [68]. Also, the functional prediction suggests sulfate oxidizing bacteria were involved in several biogeochemical cycles in the mangrove ecosystem including sulfur, nitrogen, methane and carbon metabolism. Key bacterial family like *Syntrophobacteraceae* are known to possess genes (*Apr/Dsr*) for dissimilatory sulfate reduction to $H_2S$ and are usually found to be among dominant families in sulfate rich sediments [69]. *Blastopirellula* have been implicated in anaerobic oxidation of ammonium [70]. Also, *Marinobacterium* is an established nitrogen fixing bacteria [71] while the family *Desulfobacteraceae* are versatile sulfate reducers [72] capable of coupled nitrogen metabolism [54, 73] and hydrocarbon degradation in marine environments [72, 74].

Diverse members of bacterial communities contributed differentially to carbon fixation among mangrove species. Also, predicted pathways for both nitrogen, sulfur and carbon metabolism was significantly higher in Gazi Bay compared to Mida Creek (Fig 5). This finding suggests that the inverse association of bacterial diversity and richness with the influx of nutrients, may have impacted the functional potential of the bacterial communities in Mida Creek. Further insights on genes associated with carbon metabolism revealed differential enrichment in *R. mucronata* rhizosphere. The preferential abundance of genes for carbon metabolism in mangrove sediments has been reported previously [54, 75]. Also, Zhao et at. [75] reported that total organic carbon in marine environment significantly affects the distribution of genes for carbon metabolism. For this present study, we consider that the characteristic feature of the attachment of debris to the long roots of *R. mucronata* may have influenced the bacterial community response to carbon metabolism. Overall, the site and mangrove species-based differences in predicted enzymes for energy metabolism is more likely to be a function of differences in co-interacting physicochemical parameters and critical mangrove species requirements. Though the functional prediction gives a first overview of the potential

contribution of microbial communities to energy metabolism and biosynthesis of biomolecules within mangrove rhizospheres, it is recommended that a multi-omics (metagenomics, metatranscriptomics, metabolomics and proteomics) approach, which eliminates known biases associated with PICRUSt2 prediction be applied in future explorations.

## Conclusion

This study has generated baseline data of bacterial diversity associated with rhizospheres of specific mangrove species from Kenya. The study revealed that mangrove species and interacting physicochemical terms imposes a strong pattern on the bacterial community structure and diversity of Mida Creek and Gazi Bay. The influx of nutrients into the mangrove environment inversely correlated with bacterial diversity and richness. PICRUSt2-based inference of the potential functional impact of such negative association suggested that critical pathways for sulfur and carbon metabolism could be potentially affected. However, the wide distribution of predicted genes associated with energy metabolism and the synthesis of important secondary metabolites suggest that the investigated mangroves are potentially rich hubs for the recovery of novel bacterial strains / products with wider biotechnological applications.

## Supporting information

**S1 Fig. Rarefaction curve showing sequencing depth for all the samples.**
(PDF)

**S2 Fig.** Venn diagram of shared OTUs between the same mangrove species in different sites (A) and overall shared OTUs across both study sites (B).
(PDF)

**S3 Fig.** Alpha diversity differentiation based on site (A), depth (B) and mangrove species (C).
(PDF)

**S4 Fig. Abundance and distribution of bacterial families among mangrove species in Gazi Bay and Mida Creek.**
(PDF)

**S1 Table. Significant alpha diversity correlations to physicochemical factors.**
(PDF)

**S2 Table. Test for differentially abundant genera based on depth of sampling.**
(PDF)

**S3 Table. Test of significance in multivariate space using PERMANOVA.**
(PDF)

**S4 Table. Pairwise comparison of PERMANOVA based on mangrove species.**
(PDF)

**S5 Table. Spearman's rank correlation coefficient between top bacteria genera and environmental factors.**
(PDF)

**S6 Table. Complete list of biomarkers for energy metabolism and the biosynthesis of secondary metabolites detected among mangrove species in Gazi Bay.**
(PDF)

**S7 Table. Complete list of biomarkers for energy metabolism and the biosynthesis of secondary metabolites detected among mangrove species in Mida Creek.**
(PDF)

## Acknowledgments

The NACOSTI, NEMA, KWS and KEPHIS are acknowledged for approving the research study and providing permits that facilitated field studies and shipment of samples to Canada. We also acknowledge the Centre for Forest Research and Institute for Systems and Integrative Biology of Laval University, Canada and Pwani University for support during the project.

## Author Contributions

**Conceptualization:** Edith M. Muwawa, Huxley M. Makonde, Joyce M. Jefwa, James H. P. Kahindi, Damase P. Khasa.

**Data curation:** Edith M. Muwawa, Chinedu C. Obieze.

**Formal analysis:** Edith M. Muwawa, Chinedu C. Obieze, Huxley M. Makonde.

**Funding acquisition:** Edith M. Muwawa, Huxley M. Makonde, Damase P. Khasa.

**Methodology:** Edith M. Muwawa, Huxley M. Makonde, Joyce M. Jefwa, James H. P. Kahindi, Damase P. Khasa.

**Resources:** Huxley M. Makonde, Damase P. Khasa.

**Software:** Chinedu C. Obieze.

**Supervision:** Joyce M. Jefwa, James H. P. Kahindi, Damase P. Khasa.

**Writing – original draft:** Edith M. Muwawa, Chinedu C. Obieze, Huxley M. Makonde.

**Writing – review & editing:** Edith M. Muwawa, Chinedu C. Obieze, Huxley M. Makonde, Joyce M. Jefwa, James H. P. Kahindi, Damase P. Khasa.

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
