## [Decision Letter · Decision Letter 0]

27 Aug 2020

PONE-D-20-21986

16S rRNA gene amplicon-based metagenomic analysis of bacterial communities in the rhizospheres of selected mangrove species from Mida Creek and Gazi Bay, Kenya

PLOS ONE

Dear Dr. Muwawa,

Thank you for submitting your manuscript to PLOS ONE. After careful consideration, we feel that it has merit but does not fully meet PLOS ONE’s publication criteria as it currently stands. Therefore, we invite you to submit a revised version of the manuscript that addresses the points raised during the review process.

In particular the reviewers have identified major flaws in the experimental design and the analysis that need to be addressed before considering this contribution for publication. I therefore encourage the authors to provide a revised version that address all the issues detected by the reviewers.

We look forward to receiving your revised manuscript.

Kind regards,

Marco Fusi

Academic Editor

PLOS ONE

Journal Requirements:

2. We note that you are reporting an analysis of a microarray, next-generation sequencing, or deep sequencing data set. PLOS requires that authors comply with field-specific standards for preparation, recording, and deposition of data in repositories appropriate to their field. Please upload these data to a stable, public repository (such as ArrayExpress, Gene Expression Omnibus (GEO), DNA Data Bank of Japan (DDBJ), NCBI GenBank, NCBI Sequence Read Archive, or EMBL Nucleotide Sequence Database (ENA)). In your revised cover letter, please provide the relevant accession numbers that may be used to access these data. For a full list of recommended repositories, see http://journals.plos.org/plosone/s/data-availability#loc-omics or http://journals.plos.org/plosone/s/data-availability#loc-sequencing.

3.  We note that [Figure 1] in your submission contain [map/satellite] images which may be copyrighted. All PLOS content is published under the Creative Commons Attribution License (CC BY 4.0), which means that the manuscript, images, and Supporting Information files will be freely available online, and any third party is permitted to access, download, copy, distribute, and use these materials in any way, even commercially, with proper attribution. For these reasons, we cannot publish previously copyrighted maps or satellite images created using proprietary data, such as Google software (Google Maps, Street View, and Earth). For more information, see our copyright guidelines: http://journals.plos.org/plosone/s/licenses-and-copyright.

1.     You may seek permission from the original copyright holder of Figure [1] to publish the content specifically under the CC BY 4.0 license.  

Reviewers' comments:

Reviewer's Responses to Questions

**Comments to the Author**

1. Is the manuscript technically sound, and do the data support the conclusions?

Reviewer #1: Partly

Reviewer #2: Partly

Reviewer #3: Yes

2. Has the statistical analysis been performed appropriately and rigorously? 

Reviewer #1: Yes

Reviewer #2: Yes

Reviewer #3: I Don't Know

3. Have the authors made all data underlying the findings in their manuscript fully available?

Reviewer #1: Yes

Reviewer #2: Yes

Reviewer #3: Yes

4. Is the manuscript presented in an intelligible fashion and written in standard English?

Reviewer #1: Yes

Reviewer #2: Yes

Reviewer #3: Yes

5. Review Comments to the Author

Reviewer #1: Overall the study is well presented and a lot of work has been done. However there is a major flaw in the experimental design, authors had only collected (multiple samples) only from one tree for each species so that the results cannot be generalized in any way.

Revised the text a lot of hyperbole example myriad of bacterial communities and repetition e.g. mangrove ecosystem ( 3 times in 5 lines 64-68)

Introduction: lines 70 -79 Authors refer to another studies but they simply report the main conclusion without any explanation on what they expect to find based on this results or how this study will help in their own one.

Methods:

Number of samples: even we the authors try to maximize the replication within the sampling campaign having only one tree for specie lead to a high uncertainty of the conclusion. Peculiar condition, maybe not notable, could been presented and the results cannot be generalized, it required to have more than one tree per specie sampled to corroborate the results.

How common are each species in the two sites? one is dominant and the remaining 3 rare or they shared the same % across the mangrove? please provide this data

Fig. 2 color should not be shared between phylum and genus graph

Results: Authors refer to data as no difference in depth is present for almost all the entire Results section with the exeption of the RDA results in which the differential depth is considered. If the authors want to include this factor also the previous section should include (as supplementary or how they prefer) this differentiation and data should be analyzed accordingly.

Fig. 5 need to be improved hardly readable

Discussion:

Authors refer across the text to their results without including reference to figure or table please amend. A big part of the discussion is devoted to a sort of list of which organism had been found but without adding any other information or discussion point, this part of the text should be cut or more discussion point should be inserted or merged with the following section about the putative pathway in microbial communities.

Reviewer #2: In this manuscript, Muwawa and collaborators investigated the bacterial community associated to the rhizosphere of four different mangrove species, in two distinct locations, along the coast of Kenya. The two locations present a different level of anthropogenic pollution and among these sites the authors investigated different factors that can affect the bacterial community composition associated to the mangrove sediments, highlighting the mangrove species and the chemical properties of the sediments as important factors that determine the bacterial community composition in the mangrove ecosystem.

I found this work interesting and well written. The authors through a combination of high-throughput sequencing data and sediment chemical analysis provided results that suggest how the root system of each mangrove species is important in determining the diversity and the composition of bacterial communities associated to mangrove sediments. However, I found some aspects that may require more information to strengthen the authors findings. I will report my comments in the next paragraphs.

Major issues

1. While I am convinced about the species-specific effect of route exudate on the bacterial community composition associated to the mangrove rhizosphere, I have found along the paper some contradictory parts, especially in the discussion paragraph, between the importance of location and nutrients concentration. The authors showed that the geographical location significantly affected the diversity of the bacterial community, also if it is not the main affecting factor. Authors also stated the importance of the chemical composition of the sediments, since Mida creek is characterized by a higher concentration of nutrients that decrease the bacterial diversity, due to more intense anthropogenic activities.

If it is possible, to clarify and better investigated which factors are more affecting the bacterial community composition, I think that the authors should consider to include in the analysis bulk soil samples as control, collected far from the plant roots in the two sites, to have a comparison of the bacterial community and the chemical composition associated to the sediment not affected by the mangrove root system. In this way it should be possible to consider if there are variations in the bacterial community composition that are due to differences in the sediment characteristics, that are not influenced by the presence of the different mangroves species and which parameters could be affected by anthropogenic activities.

2a. In general, I think that the authors should give more information regarding some aspects of the experiments, some also linked to my previous comment. The authors initially explained that the two sites along the Kenyan coast have been chosen due to the different level of pollution in the two areas due to anthropogenic activities, but they never mentioned which are the main parameters that could be affected in the mangrove ecosystems. Moreover, when the authors compared the two sites, they described them using different parameters. For instance, for Mida creek they provided information on the temperature range and the range of rain precipitation, while for Gazi Bay they provided the average temperature and humidity. If it is possible, I suggest that the parameters adopted to described the areas would be the same (line 110-117).

2b. In line 117-121 Authors gave a general overview on how the four mangrove species are distributed in Kenyan mangrove forest, but they don’t provide details on the distribution that each species covers in the two studied areas. Is the distribution and the proportion of the four species different in the two locations? Where the different three were collected? in the seaward side or in the middle zone? Was the tide affecting differently the mangroves sampled in the two location or at the different depths considered?

Maybe the authors could provide more information on the experimental design to better describe the mangroves in the two sampling sites and exclude that other environmental factors could have affected the bacterial community composition in the rhizosphere beside the species selection effect of the plants.

2c. Were the temperature and tide level homogeneous in the sampling sites for all the samples or there were significant variations in these parameters? These variables are usually very important and can deeply affect the bacterial community composition in mangrove sediments. May the authors comment if these parameters have been taken in consideration?

3. The functional predictions of the mangrove sediment bacterial community from 16S rRNA amplicon data has to be consider as a first overview of the bacterial community metabolic potential. The data obtained are good for the purpose of this manuscript but further studies need to investigate more in details these results, in future works. Due to the low resolution at low taxonomic level of amplicon data, the prediction could be affected by substantial bias that metagenomic or metatranscriptomic analysis can resolve.

Minor issues

1. In Figure S2 to highlight the importance of the plants on the bacterial community composition in the rhizosphere, could be more useful to see how many OTUs are in common between the same mangrove species in the 2 different locations. May the author add a Venn diagram panel also with this comparison?

I also found strange that Gazy bay does not have any OTU not shared with Mida creek. The addition of this panel may immediately give an idea on how many OTUs are shared between the rhizosphere of the same mangrove species in the two different locations, strengthen the importance of the species-specific plant effect in the selection of bacterial communities associated to the rhizosphere independently by the geographical location.

2. Probably in line 241 the authors are referring to Figure S3 panel A instead C.

3. In Figure 2 panel B I would suggest to align in the graph the relative abundance of the genus less abundant than 5%. In this way would be easier for the reader to appreciate the differences between the genus that the authors took in consideration in the analysis.

4. I suggest to add the taxonomic bacterial composition at family level in the supplementary material since authors described these results between line 279 and 293.

5. In Figure 6 change the color code for the sites, black and purple are difficult to be distinguished.

6. I found the paragraph from line 510 in the discussion redundant with the paragraph in the results section starting at line 265. I suggest that the authors rephrase the discussion paragraph avoiding the detail of the percentage already reported in the result section and focus more they comment more on their findings.

Reviewer #3: General

The work is a very interesting contribute. I recommend authors to play more attention to the clarity of sentences and writing errors. Data supports authors’ discussion and conclusions. Discussion could be more rich especially in exploring diversities in microbial communities composition in the two sites (playing attention to role of pollution that is the main difference between the two sites described by authors since the very beginning) or among different species.

Minor review

Note 1: check if measure unit and negative signs are written beside numerical data with appropriate space (especially lines 113-134).

Lines 70-78 : check for some errors in the language.

Lines 70: I’d suggest to summarize a little bit more in details the ref (10) experimental design. The sense of sentence at line 74-75 is not clear.

Lines 82-83: the phrase “due to inadequate efforts spent in exploring the mangrove habitats for microbial diversity” is maybe too strong (especially without reference).

Lines 94: better describe what kind of pollution resources are present in Mida Creek site.

Lines 109-116: data of temperature and humidity are not described homogeneously for the two sites.

Lines 138-141: method should be described more in details or a reference should be present.

Lines 452-54 are the exact copy of lines 57-58.

Lines 500-524: check for typos and errors in language.

Major review

For the description of a proper vertical distribution I’d suggest to the authors to sample at 3 different depths instead of two (done in Wu et al. 2016 – ref 10).

6. PLOS authors have the option to publish the peer review history of their article (what does this mean?). If published, this will include your full peer review and any attached files.

Reviewer #1: No

Reviewer #2: **Yes: **Alan Barozzi

Reviewer #3: No

---

## [Author Response · Author response to Decision Letter 0]

9 Oct 2020

Report on the responses raised by reviewers

PONE-D-20-21986

16S rRNA gene amplicon-based metagenomic analysis of bacterial communities in the rhizospheres of selected mangrove species from Mida Creek and Gazi Bay, Kenya

NB: Sections, page numbers and line numbers mentioned below refers to the sections, page numbers and line numbers of the ‘revised manuscript with track changes.’

Responses to comments raised by Reviewer #1: 

1. Overall the study is well presented and a lot of work has been done. However, there is a major flaw in the experimental design, authors had only collected (multiple samples) only from one tree for each species so that the results cannot be generalized in any way.

Based on background information regarding mangrove species – bacterial association, significant shifts are not expected for the same mangrove species within the same location. While we agree with the reviewer that our study would have been more robust with more sampling effort covering more trees for each mangrove specie, we understand that significant insight can still be drawn from our study with the current sampling campaign. 

2. Revised the text a lot of hyperbole example myriad of bacterial communities and repetition e.g. mangrove ecosystem (3 times in 5 lines 64-68)

We have revised the text as highlighted in the corrected manuscript (Lines 72 – 75).

3. Introduction: lines 70 -79 Authors refer to another studies but they simply report the main conclusion without any explanation on what they expect to find based on this results or how this study will help in their own one.

We have revised the section to connect its importance to our study (Lines 72 – 90).

Methods:

4. Number of samples: even we the authors try to maximize the replication within the sampling campaign having only one tree for specie lead to a high uncertainty of the conclusion. Peculiar condition, maybe not notable, could been presented and the results cannot be generalized, it required to have more than one tree per specie sampled to corroborate the results.

Sampling more trees for each mangrove specie would have certainly made the study more robust, however, the sampling effort undertaken is sufficient to provide substantial insights considering that a drastic shift is not expected within identical mangrove species in the same location. 

5. How common are each species in the two sites? one is dominant and the remaining 3 rare or they shared the same % across the mangrove? please provide this data.

We have indicated on the methodology that the predominant mangrove species contribute to over 80% of the mangrove formation in both sites (Lines 140 – 142).

6. Fig. 2 colour should not be shared between phylum and genus graph

• Changes have been made to Fig. 2 (now Fig 1) as recommended

7. Results: Authors refer to data as no difference in depth is present for almost all the entire Results section with the exeption of the RDA results in which the differential depth is considered. If the authors want to include this factor also the previous section should include (as supplementary or how they prefer) this differentiation and data should be analyzed accordingly.

• Result of depth analysis have been included in supplementary materials (S2 Table)

8. Fig. 5 need to be improved hardly readable

• Fig. 5 (now Fig 4) has been improved as recommended

9. Discussion:

Authors refer across the text to their results without including reference to figure or table please amend. A big part of the discussion is devoted to a sort of list of which organism had been found but without adding any other information or discussion point, this part of the text should be cut or more discussion point should be inserted or merged with the following section about the putative pathway in microbial communities.

• Reference to figures and Table has been included in the discussion section of the manuscript (Lines 533, 536, 547, 549, 558, 577, 606, 621)

• A large chunk of the discussion section describing organisms that have been found in mangrove environment has been deleted as recommended (Line 595-596, 606-612)

Responses to comments raised by Reviewer #2: 

In this manuscript, Muwawa and collaborators investigated the bacterial community associated to the rhizosphere of four different mangrove species, in two distinct locations, along the coast of Kenya. The two locations present a different level of anthropogenic pollution and among these sites the authors investigated different factors that can affect the bacterial community composition associated to the mangrove sediments, highlighting the mangrove species and the chemical properties of the sediments as important factors that determine the bacterial community composition in the mangrove ecosystem. I found this work interesting and well written. The authors through a combination of high-throughput sequencing data and sediment chemical analysis provided results that suggest how the root system of each mangrove species is important in determining the diversity and the composition of bacterial communities associated to mangrove sediments. However, I found some aspects that may require more information to strengthen the authors findings. I will report my comments in the next paragraphs.

We appreciate the positive comments from the reviewer.

Major issues

1. While I am convinced about the species-specific effect of route exudate on the bacterial community composition associated to the mangrove rhizosphere, I have found along the paper some contradictory parts, especially in the discussion paragraph, between the importance of location and nutrients concentration. The authors showed that the geographical location significantly affected the diversity of the bacterial community, also if it is not the main affecting factor. Authors also stated the importance of the chemical composition of the sediments, since Mida creek is characterized by a higher concentration of nutrients that decrease the bacterial diversity, due to more intense anthropogenic activities. If it is possible, to clarify and better investigated which factors are more affecting the bacterial community composition, I think that the authors should consider to include in the analysis bulk soil samples as control, collected far from the plant roots in the two sites, to have a comparison of the bacterial community and the chemical composition associated to the sediment not affected by the mangrove root system. In this way it should be possible to consider if there are variations in the bacterial community composition that are due to differences in the sediment characteristics, that are not influenced by the presence of the different mangroves species and which parameters could be affected by anthropogenic activities.

We agree with the reviewer comments on clarifying the parameters that influence the bacterial community composition within the mangroves. However, the suggestion stated (analysis of bulk soil samples as control) was not considered during the study design, hence the bulk soil samples were not analyzed. This can be considered in future comprehensive mangrove microbial studies.

Contradiction on the importance of geolocation and physicochemical parameters have been addressed (Lines 533 – 534 and 547 – 556). Specifically, statements indicating that geolocation did not influence bacterial alpha diversity have been modified. These two parameters are intertwined since the main differences between the two study sites is the higher nutrient influx in Mida creek due to anthropogenic activities. Besides, there were significant differences in alpha diversity between same mangrove species found in the different sites (Table 1)

2. (a). In general, I think that the authors should give more information regarding some aspects of the experiments, some also linked to my previous comment. The authors initially explained that the two sites along the Kenyan coast have been chosen due to the different level of pollution in the two areas due to anthropogenic activities, but they never mentioned which are the main parameters that could be affected in the mangrove ecosystems. Moreover, when the authors compared the two sites, they described them using different parameters. For instance, for Mida creek they provided information on the temperature range and the range of rain precipitation, while for Gazi Bay they provided the average temperature and humidity. If it is possible, I suggest that the parameters adopted to described the areas would be the same (line 110-117).

We have made some changes in the text and described the parameters (temperature and humidity) in both sites so that there is uniformity (Lines 124 – 139). The anthropogenic activities that are prevalent at Mida creek site have been described and are thought to cause pollution in the area that may influence the bacterial community composition (Lines 103 – 106, 130-133).

(b) In line 117-121 Authors gave a general overview on how the four mangrove species are distributed in Kenyan mangrove forest, but they don’t provide details on the distribution that each species covers in the two studied areas. Is the distribution and the proportion of the four species different in the two locations? Where the different three were collected? in the seaward side or in the middle zone? Was the tide affecting differently the mangroves sampled in the two location or at the different depths considered? Maybe the authors could provide more information on the experimental design to better describe the mangroves in the two sampling sites and exclude that other environmental factors could have affected the bacterial community composition in the rhizosphere beside the species selection effect of the plants.

We have made some changes in the methodology to describe the mangrove zonation (distribution of the mangrove species, especially the predominant species) which takes the same pattern in both sites as indicated in the text. The details of mangrove species distribution whether seaward, in the middle or landward is indicated (Lines 140 - 153).

We have also pointed out in the text the temperature and tides conditions during sampling as highlighted in the manuscript text (Lines 160-161).

(c). Were the temperature and tide level homogeneous in the sampling sites for all the samples or there were significant variations in these parameters? These variables are usually very important and can deeply affect the bacterial community composition in mangrove sediments. May the authors comment if these parameters have been taken in consideration?

We have indicated in the methodology the conditions during our sampling in the two sites. There was no significant variation in the parameters (temperature and tides) stated (Lines 160 – 161). Sampling was done during low tides at both sites. Mida creek had a temperature of 27°C, while Gazi bay had 28°C.

3. The functional predictions of the mangrove sediment bacterial community from 16S rRNA amplicon data has to be consider as a first overview of the bacterial community metabolic potential. The data obtained are good for the purpose of this manuscript but further studies need to investigate more in details these results, in future works. Due to the low resolution at low taxonomic level of amplicon data, the prediction could be affected by substantial bias that metagenomic or metatranscriptomic analysis can resolve.

We agree with the reviewer that the functional prediction is a first overview and have thus recommended the application multi-omics techniques in future investigations (Lines 669 – 676)

Minor issues

1. In Figure S2 to highlight the importance of the plants on the bacterial community composition in the rhizosphere, could be more useful to see how many OTUs are in common between the same mangrove species in the 2 different locations. May the author add a Venn diagram panel also with this comparison?

I also found strange that Gazy bay does not have any OTU not shared with Mida creek. The addition of this panel may immediately give an idea on how many OTUs are shared between the rhizosphere of the same mangrove species in the two different locations, strengthen the importance of the species-specific plant effect in the selection of bacterial communities associated to the rhizosphere independently by the geographical location.

• A venn diagram panel comparing core OTUs in each mangrove species in both sites have been included as recommended (S2 Fig). The relevant results section has also been updated (Lines 264-271)

2. Probably in line 241 the authors are referring to Figure S3 panel A instead C.

• Yes, we were referring to figure S3A. The required correction has been affected in the revised manuscript (Lines 280-285)

3. In Figure 2 panel B I would suggest to align in the graph the relative abundance of the genus less abundant than 5%. In this way would be easier for the reader to appreciate the differences between the genus that the authors took in consideration in the analysis.

• The percentage of less abundant genera have been included to the legend of Figure 2B (now Fig 1B) (Line 328 – 330).

4. I suggest to add the taxonomic bacterial composition at family level in the supplementary material since authors described these results between line 279 and 293.

• The bacterial composition at family level has been added to supplementary materials (S4 Fig)

5. In Figure 6 change the color code for the sites, black and purple are difficult to be distinguished.

• The colour code for sites has been changed as recommended (now Fig 5)

6. I found the paragraph from line 510 in the discussion redundant with the paragraph in the results section starting at line 265. I suggest that the authors rephrase the discussion paragraph avoiding the detail of the percentage already reported in the result section and focus more they comment more on their findings.

We have rephrased the discussion paragraph and removed the percentages as suggested by the reviewer. See the highlighted text in the corrected manuscript.

Responses to comments raised by Reviewer #3:

1. The work is a very interesting contribute. I recommend authors to play more attention to the clarity of sentences and writing errors. Data supports authors’ discussion and conclusions. Discussion could be more rich especially in exploring diversities in microbial communities composition in the two sites (playing attention to role of pollution that is the main difference between the two sites described by authors since the very beginning) or among different species.

We have made changes on the clarity of sentences and some grammatical errors as highlighted in the text.

Differentially abundant bacterial taxa of same mangrove species in both sites have been included in the results section (Lines 318 – 325, 352 – 369) and the corresponding discussion section has been updated (Lines 635-639)

Minor review

2. Note 1: check if measure unit and negative signs are written beside numerical data with appropriate space (especially lines 113-134).

We have confirmed that measure units and negative signs are written beside numerical data with appropriate space.

3. Lines 70-78: check for some errors in the language.

We have corrected lines 70-81 as highlighted in the text.

4. Lines 70: I’d suggest to summarize a little bit more in details the ref (10) experimental design. 

We have summarized the statement and made it clear.

5. The sense of sentence at line 74-75 is not clear.

We have corrected the entire paragraph.

6. Lines 82-83: the phrase “due to inadequate efforts spent in exploring the mangrove habitats for microbial diversity” is maybe too strong (especially without reference).

We have corrected the phrase as highlighted and given some references i.e. …... ‘However, data on microbial community diversity is limited due to low attention in exploring the mangrove habitats for microbial diversity (14,15)’

7. Lines 94: better describe what kind of pollution resources are present in Mida Creek site.

We have stated the anthropogenic activities that are prevalent at the Mida creek (Lines 103 – 133 and 130-133).

8. Lines 109-116: data of temperature and humidity are not described homogeneously for the two sites.

We have corrected the referred lines in the study site section and provided homogeneous data of temperature and humidity as highlighted in the text (Lines 125-139).

9. Lines 138-141: method should be described more in details or a reference should be present.

We have provided references for the methods used as highlighted in the text (Lines 173 – 176).

10. Lines 452-54 are the exact copy of lines 57-58.

We have corrected the referred statements in the lines (520 - 522) under the discussion section as highlighted in the text.

11. Lines 500-524: check for typos and errors in language.

We have corrected the typos and errors in the referred text section as indicated in the text.

Major review

12. For the description of a proper vertical distribution I’d suggest to the authors to sample at 3 different depths instead of two (done in Wu et al. 2016 – ref 10).

We agree with the reviewer’s comment; however, we did not factor that in during our study design, partly due to some limited funding and we can recommend for such design in a future comprehensive study on mangrove microbial communities.

Response to Editor’s comments:

Journal Requirements:

We have ensured that all the PLOS One’s style requirements have been adhered to thorough the manuscript.

2. We note that you are reporting an analysis of a microarray, next-generation sequencing, or deep sequencing data set. PLOS requires that authors comply with field-specific standards for preparation, recording, and deposition of data in repositories appropriate to their field. Please upload these data to a stable, public repository (such as ArrayExpress, Gene Expression Omnibus (GEO), DNA Data Bank of Japan (DDBJ), NCBI GenBank, NCBI Sequence Read Archive, or EMBL Nucleotide Sequence Database (ENA)). In your revised cover letter, please provide the relevant accession numbers that may be used to access these data. For a full list of recommended repositories, see http://journals.plos.org/plosone/s/data-availability#loc-omics or http://journals.plos.org/plosone/s/data-availability#loc-sequencing.

Our sequencing data have been uploaded onto the NCBI Sequence Read Archi (Ref: BioProject ID PRJNA644929).

3. We note that [Figure 1] in your submission contain [map/satellite] images which may be copyrighted. All PLOS content is published under the Creative Commons Attribution License (CC BY 4.0), which means that the manuscript, images, and Supporting Information files will be freely available online, and any third party is permitted to access, download, copy, distribute, and use these materials in any way, even commercially, with proper attribution. For these reasons, we cannot publish previously copyrighted maps or satellite images created using proprietary data, such as Google software (Google Maps, Street View, and Earth). For more information, see our copyright guidelines: http://journals.plos.org/plosone/s/licenses-and-copyright.

We have removed the figure to avoid the copyright issues.

---

## [Decision Letter · Decision Letter 1]

8 Dec 2020

PONE-D-20-21986R1

16S rRNA gene amplicon-based metagenomic analysis  of bacterial communities in the rhizospheres of selected mangrove species from Mida Creek and Gazi Bay, Kenya

PLOS ONE

Dear Dr. Muwawa,

Thank you for submitting your manuscript to PLOS ONE. After careful consideration, we feel that it has merit but does not fully meet PLOS ONE’s publication criteria as it currently stands. Therefore, we invite you to submit a revised version of the manuscript that addresses the points raised during the review process.

Although the manuscript was significantly improved, the author still need to provide a clear explanation of the term rhizosphere. The revised version provides an explanation of the methods use to retrieve the rhizopsheric soil that does not match with a rigorous definition of rhizosphere. Rhizosphere is that portion of sediment attached to the root. Specific procedures are needed to separate the rhizosphere from the root from the sediment surrounding the root. See the comment of the reviewer 4 for more detail. If unable to provide such methods the authors can think to remodel the title and the main message of the paper, that is still valuable especially in ecosystems such as mangroves where microbial studies are poor.

We look forward to receiving your revised manuscript.

Kind regards,

Marco Fusi

Academic Editor

PLOS ONE

Reviewers' comments:

Reviewer's Responses to Questions

**Comments to the Author**

1. If the authors have adequately addressed your comments raised in a previous round of review and you feel that this manuscript is now acceptable for publication, you may indicate that here to bypass the “Comments to the Author” section, enter your conflict of interest statement in the “Confidential to Editor” section, and submit your "Accept" recommendation.

Reviewer #2: (No Response)

Reviewer #3: All comments have been addressed

Reviewer #4: (No Response)

2. Is the manuscript technically sound, and do the data support the conclusions?

Reviewer #2: Yes

Reviewer #3: Yes

Reviewer #4: Yes

3. Has the statistical analysis been performed appropriately and rigorously? 

Reviewer #2: Yes

Reviewer #3: Yes

Reviewer #4: Yes

4. Have the authors made all data underlying the findings in their manuscript fully available?

Reviewer #2: Yes

Reviewer #3: Yes

Reviewer #4: Yes

5. Is the manuscript presented in an intelligible fashion and written in standard English?

Reviewer #2: Yes

Reviewer #3: Yes

Reviewer #4: No

6. Review Comments to the Author

Reviewer #2: I found the revised version of the manuscript of Muwawa and collaborators clearly improved. The authors put a good effort to comply with the reviewers’ comments and I do not have further clarification to ask to the authors. However, I reported a very few minor comments, that I think need to be addressed before the publication.

Minor comments

- In the abstract line 27, Mida Creek and Gazi Bay are both in capital letters, while in the manuscript after this point, the capital letter was removed in the second part of the name. In line 112 happened the same. Give a check and be consistent.

- In lines 50-52, in the brackets there is a space between the numbers and the percentage symbol. I suggest to remove it and in general to keep the same style along the all text (i.e. in line 52 there is no space).

- In lines 125-126 “Gazy bay and Mida creek had low tides and temperature of 28°C and 27°C.” Are the authors referring at the conditions found at the time of the sampling?

Remove this sentence here, since it is out of context and it is repeated already in the collection of samples paragraph, where gives important information on the conditions in the sampling sites.

- In Figure 1 panel B, (old Figure 2) my comment was misunderstood. If it is possible, I suggested to align in the graph, in panel B, the relative abundance of the genus less abundant than 4%. In this way it would be easier for the reader to appreciate the differences between the genus that the authors took in consideration in the analysis and there will not be some taxa before and after the “others” bar, like the author did in panel A.

- Maybe it is my mistake but I didn’t fine the Supplementary Figure S4 mentioned in line 313.

- In line 351 probably the authors are referring to figure 2 and not 3.

- For Figure 4, there is only the caption title. A small description of the figure could help the reader to understand it.

- Lines 545-548, the two sentences are quite repetitive. Try to rephrase them better.

Reviewer #3: In my opinion the corrections applied are adequate.

The paper is now more clear and precise in form and contents.

Line 67-69: check the sentence grammar.

Reviewer #4: General:

In the manuscript titled “16S rRNA gene amplicon-based metagenomic analysis of bacterial communities in the rhizospheres of selected mangrove species from Mida Creek and Gazi Bay, Kenya” by Muwawa et al., the rhizosphere of four different mangrove species at two sites along the coast of Kenya is analysed for sedimentary conditions and the microbial community composition. The study shows how different areas within the forest dominated by different tree species contain varying bacterial communities. Furthermore, the authors identify potential drivers of bacterial community composition. This work provides novel insights into the bacterial communities of mangrove soils in eastern Africa, and adds to a growing datatbase of mangrove microbiomes across the globe. I commend the authors for their work and the neat presentation of the manuscript. I recommend however, to complete a thorough grammar-, spell-, and punctuation-check on the entire document as there were too many mistakes for me to point out.

I do not agree with the definition of rhizospheric sediment as it is presented in the manuscript, as it seems that the samples were taken from below the trees and with roots present in the sample, but there is no evidence that this area is actively mediated by root exudates (I would have sampled the soil directly off the roots). As there may be wider definitions of what the rhizosphere is (I am aware the methodology followed Wu et al. 2016), I would like to see it defined for this study, and the sampling method justified.

Throughout the document, there is an inconsistency using R. mucronate instead of R. mucronata

Abstract:

There is no need for specification of methods (such as which alpha diversity metrics used) and p-values in the abstract

Introduction:

Lines 125-126: This sentence gets repeated in the “collection of samples” section. Can be removed here.

Lines 126-133: The contribution of each species is unclear: Do all four species together contribute to 80% of the mangrove forest? If yes, what is the contribution of each species? This doesn’t need to be exact, but if the microbiome varies between these stands, it would be good to know their approximate extent.

Line 142: Four trees of each species means 4x4 =16 individual trees were sampled. If this is the case, I apologise, please ignore the comment. However, I believe “one tree each of the four species was sampled”, is what was actually done.

Lines 143-145: I find the description of the sampling procedure confusing. It may help to break down the sentences as much as possible to keep each action simple and clear. A diagram of the sampling design would help immensely.

Line 146: When calculating the number of samples a 4x4x2x2 structure is used, while it is described in the text as 4x8x2. Keep it consistent for clarity.

Line 198: I wonder, why OTUs were used for community metrics but ASVs for the functional alignment? ASVs and OTUs are two methods, that rely on very different ideas. In this way, I do not think the community data and the functional data can be compared.

Results:

Line 351: Figure mention should be Fig. 2 instead of Fig. 3

Lines 394-398: The wording in these sentences doesn’t quite fit the analysis. The environmental factors are able to separate the sites/samples, ie pH separated the samples along the y axis (Gazi vs Mida). pH has an effect on the bacterial community in all sites but is here able to explain some of the differences between them.

Lines 416-420: The interpretation of the variance partitioning results is not quite accurate. The “interaction zones” (overlap of the diagram) are additive to the influence of a single factor. This means that the variation explained by sediment chemical properties alone was 6% and when an interaction with species was inferred it was 11%. Hence, the interaction doesn’t reduce the influence of the factors but adds another layer of information.

Discussion:

I would like to see a note in the discussion in which the authors discuss the potential effect of intertidal zonation on the microbiome. They show nicely how the tree species have an effect on the bacterial community, but fail to mention in the discussion, that the trees follow a strict zonation from sea to land. While the study is not designed to disentangle the question whether the trees or their intertidal position determine the microbiome more, this is an important point to mention. Especially as the sedimentary conditions have such a large influence too.

7. PLOS authors have the option to publish the peer review history of their article (what does this mean?). If published, this will include your full peer review and any attached files.

Reviewer #2: **Yes: **Alan Barozzi

Reviewer #3: No

Reviewer #4: **Yes: **Timothy Thomson

---

## [Author Response · Author response to Decision Letter 1]

16 Jan 2021

Report on the responses raised by reviewers

PONE-D-20-21986

16S rRNA gene amplicon-based metagenomic analysis of bacterial communities in the rhizospheres of selected mangrove species from Mida Creek and Gazi Bay, Kenya

Responses to comments raised by Reviewer #2: 

 I found the revised version of the manuscript of Muwawa and collaborators clearly improved. The authors put a good effort to comply with the reviewers’ comments and I do not have further clarification to ask to the authors. However, I reported a very few minor comments, that I think need to be addressed before the publication.

1. In the abstract line 27, Mida Creek and Gazi Bay are both in capital letters, while in the manuscript after this point, the capital letter was removed in the second part of the name. In line 112 happened the same. Give a check and be consistent.

Mida Creek and Gazi Bay is now used consistently throughout the manuscript.

2. In lines 50-52, in the brackets there is a space between the numbers and the percentage symbol. I suggest to remove it and in general to keep the same style along the all text (i.e. in line 52 there is no space).

Done as recommended (lines 49 – 51)

3. In lines 125-126 “Gazy bay and Mida creek had low tides and temperature of 28°C and 27°C.” Are the authors referring at the conditions found at the time of the sampling?

Remove this sentence here, since it is out of context and it is repeated already in the collection of samples paragraph, where gives important information on the conditions in the sampling sites.

Yes, the stated conditions are those that were found at the time of the sampling.

The sentence has been removed now.

4. In Figure 1 panel B, (old Figure 2) my comment was misunderstood. If it is possible, I suggested to align in the graph, in panel B, the relative abundance of the genus less abundant than 4%. In this way it would be easier for the reader to appreciate the differences between the genus that the authors took in consideration in the analysis and there will not be some taxa before and after the “others” bar, like the author did in panel A.

Fig 1 panel B has been replaced and recommendation by reviewer has been effected.

5. Maybe it is my mistake but I didn’t fine the Supplementary Figure S4 mentioned in line 313

Fig S4 is in line 319.

6. In line 351 probably the authors are referring to figure 2 and not 3

Authors were referring to Figure 2.

7. For Figure 4, there is only the caption title. A small description of the figure could help the reader to understand it.

A small description of the figure has been included (Lines 454 – 458)

8. Lines 545-548, the two sentences are quite repetitive. Try to rephrase them better

Done as recommended (545 – 546)

Responses to comments raised by Reviewer #3: 

In my opinion the corrections applied are adequate.

The paper is now more clear and precise in form and contents.

1. Line 67-69: check the sentence grammar.

The sentence in line 67-69 has been rephrased to correct the grammar.

Responses to comments raised by Reviewer #4:

1. In the manuscript titled “16S rRNA gene amplicon-based metagenomic analysis of bacterial communities in the rhizospheres of selected mangrove species from Mida Creek and Gazi Bay, Kenya” by Muwawa et al., the rhizosphere of four different mangrove species at two sites along the coast of Kenya is analysed for sedimentary conditions and the microbial community composition. The study shows how different areas within the forest dominated by different tree species contain varying bacterial communities. Furthermore, the authors identify potential drivers of bacterial community composition. This work provides novel insights into the bacterial communities of mangrove soils in eastern Africa, and adds to a growing datatbase of mangrove microbiomes across the globe. I commend the authors for their work and the neat presentation of the manuscript. 

We appreciate the positive comments.

I recommend however, to complete a thorough grammar-, spell-, and punctuation-check on the entire document as there were too many mistakes for me to point out.

We have revised and corrected the entire document for grammar, spelling and punctuation mistakes as recommended.

I do not agree with the definition of rhizospheric sediment as it is presented in the manuscript, as it seems that the samples were taken from below the trees and with roots present in the sample, but there is no evidence that this area is actively mediated by root exudates (I would have sampled the soil directly off the roots). As there may be wider definitions of what the rhizosphere is (I am aware the methodology followed Wu et al. 2016), I would like to see it defined for this study, and the sampling method justified. 

We have edited the sampling methods to explain the procedure of obtaining the rhizosphere sediment. The coring was performed closely to the mangrove roots in each case in order to capture the rhizosphere sediment. During the DNA extraction, the excised roots containing rhizosphere sediment were placed in a 50 mL sterile falcon tube containing autoclaved phosphate buffer and shaken for 2 min to release the rhizosphere sediment from the surface of the roots.

Throughout the document, there is an inconsistency using R. mucronate instead of R. mucronata.

The name R. mucronate has been changed to R. mucronate throughout the document.

Abstract

2. There is no need for specification of methods (such as which alpha diversity metrics used) and p-values in the abstract

Done as recommended (Lines 28 – 29, 31 – 34, 40 and 42)

Introduction

3. Lines 125-126: This sentence gets repeated in the “collection of samples” section. Can be removed here.

The sentence in lines 125-126 has been removed.

4. Lines 126-133: The contribution of each species is unclear: Do all four species together contribute to 80% of the mangrove forest? If yes, what is the contribution of each species? This doesn’t need to be exact, but if the microbiome varies between these stands, it would be good to know their approximate extent.

Yes, all the four species together contribute to 80% of the mangrove forest. The contribution of each species was not exactly determined during sampling, but it can be estimated in terms of dominance as A. marina (30%), R. Mucronate (25%), S. alba (15%) and C. tagal (10%).

5. Line 142: Four trees of each species means 4x4 =16 individual trees were sampled. If this is the case, I apologise, please ignore the comment. However, I believe “one tree each of the four species was sampled”, is what was actually done.

Line 142 was corrected to mean ‘Four mangrove tree species’ i.e. one tree for each of the four species selected.

6. Lines 143-145: I find the description of the sampling procedure confusing. It may help to break down the sentences as much as possible to keep each action simple and clear. A diagram of the sampling design would help immensely.

The sampling procedure has been revised as recommended, but we were not able to provide diagram. 

7. Line 146: When calculating the number of samples a 4x4x2x2 structure is used, while it is described in the text as 4x8x2. Keep it consistent for clarity.

The description in line 142 has been corrected to mean 4 mangrove tree species so as to be consistent with the 4x4x2x2 structure indicated. 

8. Line 198: I wonder, why OTUs were used for community metrics but ASVs for the functional alignment? ASVs and OTUs are two methods, that rely on very different ideas. In this way, I do not think the community data and the functional data can be compared.

The functional prediction has been reversed from ASVs to OTUs for uniformity (Line 200, 436 – 443, 445 - 473). Picrust2 functional prediction is based on representative sequences of either ASVs or OTUs. In our case, OTUs were clustered from ASVs and it was on this basis that representative ASV sequences was used for the prediction.

Results

9. Line 351: Figure mention should be Fig. 2 instead of Fig. 3

Done (Line 357)

10. Lines 394-398: The wording in these sentences doesn’t quite fit the analysis. The environmental factors are able to separate the sites/samples, ie pH separated the samples along the y axis (Gazi vs Mida). pH has an effect on the bacterial community in all sites but is here able to explain some of the differences between them.

The result presentation in this section has been modified to clearly reflect the influence of pH on the bacterial communities within and across study sites (Lines 400 – 402)

11. Lines 416-420: The interpretation of the variance partitioning results is not quite accurate. The “interaction zones” (overlap of the diagram) are additive to the influence of a single factor. This means that the variation explained by sediment chemical properties alone was 6% and when an interaction with species was inferred it was 11%. Hence, the interaction doesn’t reduce the influence of the factors but adds another layer of information.

The interpretation has been updated and the needed corrections effected (Lines 418 – 426)

Discussion

12. I would like to see a note in the discussion in which the authors discuss the potential effect of intertidal zonation on the microbiome. They show nicely how the tree species have an effect on the bacterial community, but fail to mention in the discussion, that the trees follow a strict zonation from sea to land. While the study is not designed to disentangle the question whether the trees or their intertidal position determine the microbiome more, this is an important point to mention. Especially as the sedimentary conditions have such a large influence too. 

The discussion section has been updated to include the potential influence of mangrove zonation from sea to land on the bacterial communities (Lines 263 – 269 and lines 550 – 557).

---

## [Decision Letter · Decision Letter 2]

11 Feb 2021

PONE-D-20-21986R2

16S rRNA gene amplicon-based metagenomic analysis  of bacterial communities in the rhizospheres of selected mangrove species from Mida Creek and Gazi Bay, Kenya

PLOS ONE

Dear Dr. Muwawa,

Thank you for submitting your manuscript to PLOS ONE. After careful consideration, we feel that it has merit but does not fully meet PLOS ONE’s publication criteria as it currently stands. Therefore, we invite you to submit a revised version of the manuscript that addresses the points raised during the review process.

The authors were able to address all the reviewer and editor points but there is still two minor changes that need to be solved as indicated by the reviewer, before to consider the manuscript for publication. 

We look forward to receiving your revised manuscript.

Kind regards,

Marco Fusi

Academic Editor

PLOS ONE

Reviewers' comments:

Reviewer's Responses to Questions

**Comments to the Author**

1. If the authors have adequately addressed your comments raised in a previous round of review and you feel that this manuscript is now acceptable for publication, you may indicate that here to bypass the “Comments to the Author” section, enter your conflict of interest statement in the “Confidential to Editor” section, and submit your "Accept" recommendation.

Reviewer #4: (No Response)

2. Is the manuscript technically sound, and do the data support the conclusions?

Reviewer #4: Yes

3. Has the statistical analysis been performed appropriately and rigorously? 

Reviewer #4: Yes

4. Have the authors made all data underlying the findings in their manuscript fully available?

Reviewer #4: Yes

5. Is the manuscript presented in an intelligible fashion and written in standard English?

Reviewer #4: Yes

6. Review Comments to the Author

Reviewer #4: I found my comments on the manuscript by Muwawa et al. addressed satisfyingly.

Two minor issues with the wording may be resolved as suggested to ensure clarity of the study site and design. If these two sentences can be fixed, I am happy to recommend this mauscript for publication.

Methods:

Line 124-127: A suggestion: The mangrove forests in both sites display a similar zonation and mangrove species contribution among the four dominant species A. marina (30%), R. mucronata (25%), S. alba (15%), and C. tagal (10%), which contribute about 80% of the total mangrove formation in both forests.

Lines 140-142: The sentences seem repetitive. You also use the word species in a ambivalent way, which makes it very hard to understand. I would suggest a very simple sentence such as: “From pure stands of S. alba, R. mucronata, C. tagal, and A. marina, one specimen (or tree) was selected per stand. The selected trees were spaced at least 10 m apart.”

7. PLOS authors have the option to publish the peer review history of their article (what does this mean?). If published, this will include your full peer review and any attached files.

Reviewer #4: **Yes: **Timothy Thomson

---

## [Author Response · Author response to Decision Letter 2]

25 Feb 2021

Report on the responses raised by reviewer #4

PONE-D-20-21986R2

16S rRNA gene amplicon-based metagenomic analysis of bacterial communities in the rhizospheres of selected mangrove species from Mida Creek and Gazi Bay, Kenya

Responses to comments raised by Reviewer #4:

I found my comments on the manuscript by Muwawa et al. addressed satisfyingly. Two minor issues with the wording may be resolved as suggested to ensure clarity of the study site and design. If these two sentences can be fixed, I am happy to recommend this mauscript for publication.

Methods:

1. Line 124-127: A suggestion: The mangrove forests in both sites display a similar zonation and mangrove species contribution among the four dominant species A. marina (30%), R. mucronata (25%), S. alba (15%), and C. tagal (10%), which contribute about 80% of the total mangrove formation in both forests.

Lines124-127 have been revised as suggested by the reviewer;

The mangrove forests in both sites display a similar zonation and mangrove species contribution among the four dominant species A. marina (30%), R. Mucronata (25%), S. alba (15%) and C. tagal (10%), which contribute about 80% of the total mangrove formation in both forests.

2. Lines 140-142: The sentences seem repetitive. You also use the word species in a ambivalent way, which makes it very hard to understand. I would suggest a very simple sentence such as: “From pure stands of S. alba, R. mucronata, C. tagal, and A. marina, one specimen (or tree) was selected per stand. The selected trees were spaced at least 10 m apart.”

Lines140-142 have been revised as recommended by the reviewer;

From pure stands of S. alba, R. mucronata, C. tagal and A. marina, one tree was selected per stand. The selected trees were spaced at least 10 m apart.

---

## [Editor Report · Decision Letter 3]

1 Mar 2021

16S rRNA gene amplicon-based metagenomic analysis  of bacterial communities in the rhizospheres of selected mangrove species from Mida Creek and Gazi Bay, Kenya

PONE-D-20-21986R3

Dear Dr. Muwawa,

We’re pleased to inform you that your manuscript has been judged scientifically suitable for publication and will be formally accepted for publication once it meets all outstanding technical requirements.

Kind regards,

Marco Fusi

Academic Editor

PLOS ONE
---

## [Editor Report · Acceptance letter]

11 Mar 2021

PONE-D-20-21986R3 

16S rRNA gene amplicon-based metagenomic analysis of bacterial communities in the rhizospheres of selected mangrove species from Mida Creek and Gazi Bay, Kenya 

Dear Dr. Muwawa:

I'm pleased to inform you that your manuscript has been deemed suitable for publication in PLOS ONE. Congratulations! Your manuscript is now with our production department. 

Kind regards, 

on behalf of

Dr. Marco Fusi 

Academic Editor

PLOS ONE